# Cross-Domain Graph Anomaly Detection via Test-Time Training with Homophily-Guided Self-Supervision

**Delaram Pirhayatifard**                                                  *dp43@rice.edu*
*Department of Electrical and Computer Engineering*
*Rice University*

**Arlei Silva**                                                            *arlei@rice.edu*
*Department of Computer Science*
*Ken Kennedy Institute for Responsible AI and Computing for Global Impact*
*Rice University*

**Reviewed on OpenReview:** *https://openreview.net/forum?id=sB3LqdOlNb*

## Abstract

Graph Anomaly Detection (GAD) has demonstrated great effectiveness in identifying unusual patterns within graph-structured data. However, while labeled anomalies are often scarce in emerging applications, existing supervised GAD approaches are either ineffective or not applicable when moved across graph domains due to distribution shifts and heterogeneous feature spaces. To address these challenges, we present GADT3, a novel test-time training framework for cross-domain GAD. GADT3 combines supervised and self-supervised learning during training while adapting to a new domain during test time using only self-supervised learning by leveraging a homophily-based affinity score that captures domain-invariant properties of anomalies. Our framework introduces four key innovations to cross-domain GAD: an effective self-supervision scheme, an attention-based mechanism that dynamically learns edge importance weights during message passing, domain-specific encoders for handling heterogeneous features, and class-aware regularization to address imbalance. Experiments across multiple cross-domain settings demonstrate that GADT3 significantly outperforms existing approaches, achieving average improvements of over 8.2% in AUROC and AUPRC compared to the best competing model. The source code for GADT3 is available at `https://github.com/delaramphf/GADT3-Algorithm`.

## 1 Introduction

Graph Anomaly Detection (GAD) is a critical task to identify unusual patterns (or outliers) in graph-structured data (Ma et al., 2021; Akoglu et al., 2015). This problem has many real-world applications, such as in e-commerce (Zhang et al., 2022), social networks (Venkatesan & Prabhavathy, 2019), fraud detection (Jiang et al., 2019), and cybersecurity (Lazarevic et al., 2003). A key limitation of existing GAD models is that they face unique cross-domain challenges that distinguish them from general graph learning tasks. These include inconsistent definitions of normal/anomalous patterns and heterogeneous feature spaces between domains (e.g., Facebook vs. Amazon datasets). These applications would benefit from GAD models capable of adapting across Out-Of-Distribution (OOD) and heterogeneous graphs.

A major motivation for our work is cybersecurity. Network intrusion detection systems play a key role in identifying malicious network activity associated with cyberattacks (Tsai et al., 2009; Kilincer et al., 2021). Due to the volume and dynamic nature of these attacks, modern intrusion detection systems increasingly rely on large amounts of data and machine learning to assist cybersecurity experts in detecting potential intrusions. However, one of the major challenges in data-driven cybersecurity is the lack of sufficient labeled data for training supervised models. This is particularly critical for emerging applications, where traces of

labeled intrusions are scarce, and the ability to leverage attacks or other types of anomalous behavior from existing labeled datasets would be greatly beneficial.

This paper investigates the unsupervised domain adaptation problem for GAD, focusing on the transfer of knowledge from a labeled source to an unlabeled target domain. While domain adaptation for graphs has been studied by previous work (Shi et al., 2024; Wang et al., 2021; Zheng et al., 2019; Yin et al., 2023; Wang et al., 2023a; Ding et al., 2018), cross-domain graph anomaly detection is an emerging challenge (Wang et al., 2023b; Ding et al., 2021). We propose using Test-Time Training (TTT) (Sun et al., 2020) to address cross-domain graph anomaly detection. Unlike traditional domain adaptation methods that require continuous access to source data, TTT allows us to distill source knowledge into model parameters and adapt to target domains using only self-supervision. This is particularly advantageous for graph anomaly detection, where source data may be sensitive or unavailable during the adaptation phase.

Our approach, Graph Anomaly Detection with Test-time Training (GADT3), has four major innovations compared to existing work on cross-domain GAD. First, it leverages a homophily-based affinity score (Qiao & Pang, 2024; Chen et al., 2024b) for self-supervised learning based on the observation that normal nodes are more similar to their neighbors than anomalous ones, a pattern that remains consistent across domains. We observe a clear separation in homophily scores between normal and anomalous nodes across most datasets, justifying the use of homophily-based SSL. Figure 1 shows the homophily score distributions for two datasets (see Appendix A.11 for detailed plots). Second, it introduces Normal Structure-preserved Attention Weighting (NSAW), which dynamically learns continuous edge importance weights through the attention mechanism to suppress the effect of anomalous nodes on the representation of normal ones. Third, GADT3 applies source and target-specific encoders that are trained end-to-end to handle both distribution shifts and heterogeneous feature spaces. Fourth, to address the extreme class imbalance challenge in GAD (Ma et al., 2024), our approach employs class-aware regularization during source training—with a stronger regularization to the minority class. By combining these innovations, GADT3 can effectively identify anomalous nodes across heterogeneous datasets arising from multiple and diverse domains.

We compare our solution against both graph domain adaptation and graph anomaly detection approaches using multiple cross-domain datasets. For instance, we show that labeled anomalies in the Amazon dataset (source) can improve the GAD accuracy on the Facebook dataset (target). The experiments show that GADT3 significantly outperforms the alternatives in most of the settings. The contributions of this paper can be summarized as:

- We propose the first test-time training (TTT) framework for cross-domain graph anomaly detection (GADT3). TTT enables adaptation to new target domains without requiring direct access to source data or target labels at test time. Our approach leverages homophily-based self-supervision, which captures universal patterns, as anomalous nodes typically exhibit lower homophily compared to normal nodes.

- Our message-passing technique introduces homophily-based attention weights that naturally suppress anomalous influences on normal nodes while preserving graph structure, enabling robust anomaly detection.

- We propose several strategies specifically designed for cross-domain GAD, such as data-specific encoders, class-aware regularization, and an early-stopping strategy to prevent overfitting during test-time adaptation.

- GADT3's performance is demonstrated through extensive experiments on cross-network tasks using multiple datasets and baselines that span both domain adaptation and anomaly detection. Our approach achieves average relative improvements of over 8.2% in AUROC and AUPRC compared to the best competing model.

## 2 Related Work

### 2.1 Graph Domain Adaptation (GDA)

While graph domain adaptation (GDA) methods have shown success in transferring knowledge across different domains (Shi et al., 2024), they face fundamental limitations when applied to anomaly detection tasks. Traditional GDA methods are designed for node classification scenarios with balanced classes and consistent label semantics.

Traditional Graph Domain Adaptation (GDA) approaches, such as GRADE (Wu et al., 2023), AdaGCN (Dai et al., 2022), and UDA-GCN (Wu et al., 2020), and spectral regularization methods (You et al., 2023), focus on minimizing distribution shifts across domains but require training samples during inference, limiting their real-time adaptation capabilities. Test-time training (TTT) (Sun et al., 2020) has emerged as a powerful framework for handling distribution shifts (Li et al., 2022; Wu et al., 2024; Zhang et al., 2024; Liu et al., 2021b). Methods like GTrans (Jin et al., 2022), GraphCL (You et al., 2020), and GT3 (Wang et al., 2022) propose graph-specific TTT approaches. TENT (Wang et al., 2020) uses prediction entropy minimization, while GraphTTA (Chen et al., 2022) leverages information theory for TTT on graphs. However, existing GDA approaches, including both traditional methods and TTT-based solutions, face key limitations for anomaly detection as they (1) don't account for extreme class imbalance within domains, (2) lack mechanisms to preserve anomaly-indicating patterns during transfer, and (3) don't handle heterogeneous feature spaces across domains. While there exists limited research on cross-domain graph anomaly detection, prior approaches require direct access to the source dataset during adaptation (Ding et al., 2021; Wang et al., 2023b). Our approach offers key advantages over these methods by eliminating the need to store and access the full source dataset during adaptation. This makes GADT3 both privacy-preserving and memory-efficient, thus being more practical for real-world deployment where these are major concerns.

Our work addresses key challenges in deploying GDA in anomaly detection through dataset-specific encoders for heterogeneous feature spaces and a homophily-based unsupervised learning approach that extracts domain-invariant properties of anomalies that generalize across domains. In our experiments, we show that GADT3 often outperforms multiple GDA baselines, including GRADE, AdaGCN, TENT, and GTrans.

### 2.2 Graph Anomaly Detection (GAD)

Graph Anomaly Detection (GAD) (Ma et al., 2021; Akoglu et al., 2015) focuses on identifying abnormal nodes in graph-structured data. Traditional approaches like Oddball (Akoglu et al., 2010) rely on power-law relationships between local graph features, while more recent deep learning-based approaches are more generalizable. For instance, DOMINANT (Ding et al., 2019) employs a graph autoencoder to identify anomalies based on graph reconstruction. ComGA (Luo et al., 2022) introduces a tailored GCN to learn distinguishable node representations by explicitly capturing community structure. Self-supervised techniques have emerged as powerful tools for GAD, with methods like CoLA (Liu et al., 2021a), SL-GAD (Zheng et al., 2021), HCM-A (Huang et al., 2022), and TAM (Qiao & Pang, 2024) introducing various approaches to handle node interactions and structural patterns. While these methods have shown success in single-domain

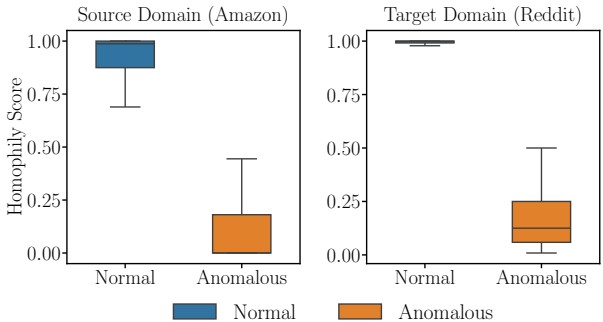

Figure 1: Homophily score distributions across domains (Amazon to Reddit): normal nodes (blue) show higher scores than anomalous nodes (orange), suggesting homophily as a domain-invariant anomaly signal.

scenarios, they don't address the challenges of cross-domain knowledge transfer. Recent methods like ACT (Wang et al., 2023b) and COMMANDER (Ding et al., 2021) have started to explore cross-domain graph anomaly detection (CD-GAD), but this remains an evolving field. Other recent works like CDFS-GAD (Chen et al., 2024a) address cross-domain node anomaly detection in a few-shot setting, where a small set of

labeled anomalies from the target graph is available. ARC (Liu et al., 2024) proposes an in-context learning approach for cross-domain GAD, which adapts at inference using a few normal nodes from the target graph.

Anomalies behave differently across domains—fraudulent users in e-commerce networks exhibit different patterns compared to social networks. Second, domains often have different feature spaces and graph structures. While our method is motivated by the homophily assumption, recent studies such as (Gao et al., 2023) have explored anomaly detection in heterophilic graphs, highlighting the importance of developing models robust to varying structural patterns. Our approach can be extended to heterophilic graphs by replacing the current SSL loss with a loss more suited for heterophily, and by using heterophily-aware GNNs.

GADT3 bridges both GDA and GAD through a unified approach that combines (1) test-time training with a homophily-based loss for capturing domain-invariant properties of anomalies (2) enhanced message-passing using normal structure-preserved attention weighting (NSAW) that reduces the irrelevant impact of anomalous nodes on normal nodes' representations, and (3) class-aware regularization that prevents minority class patterns from being overshadowed by the majority class during source training. In our experiments, we compare a source-free version of GADT3 against GAD baselines (e.g., TAM, ComGA, DOMINANT) to highlight the impact of its GAD-specific features.

## 3 Problem Definition

We address unsupervised node-level anomaly detection with domain adaptation, aiming to identify abnormal nodes in a target graph by leveraging information from both the target and a source graph. The source dataset contains labeled information but is assumed to be out-of-distribution (OOD) relative to the target, with different feature sets and label distributions. Given source dataset $D_s = (\mathcal{G}_s, X_s, Y_s)$ and target dataset $D_t = (\mathcal{G}_t, X_t)$, where $\mathcal{G}_s/\mathcal{G}_t = (\mathcal{V}, \mathcal{E})$ represents a (source or target) underlying graph with nodes $\mathcal{V}$ and edges $\mathcal{E} \subset \mathcal{V} \times \mathcal{V}$ such that $uv \in \mathcal{E}$ if there is a link between nodes $u$ and $v$. Here, $X_s = \{\mathbf{x}_v | \forall v \in \mathcal{V}_s\}$ and $Y_s = \{y_v | \forall v \in \mathcal{V}_s\}$ represent the node features and their corresponding labels, respectively, and the same holds for target data $(X_t, Y_t)$. Notably, $y_v \in \{0, 1\}$ (normal or anomaly) for nodes in both $\mathcal{V}_s$ and $\mathcal{V}_t$. Our goal is to detect anomalies in the target graph. The key challenge is that target labels are completely unavailable (unsupervised setting), requiring us to leverage source labels $Y_s$ as auxiliary information, even though source and target features may have different dimensionalities and semantic meanings ($\mathbf{x}_v \in \mathbb{R}^{p_s}$ for source nodes and $\mathbf{x}_u \in \mathbb{R}^{p_t}$ for target nodes).

## 4 Graph Anomaly Detection via Test-Time Training (GADT3)

We propose GADT3 (Graph Anomaly Detection with Test-time Training), a domain adaptation method particularly designed for graph anomaly detection that integrates attention-based message-passing, node homophily patterns, and domain-specific encoders to handle feature shifts and heterogeneous features across domains. GADT3 addresses the challenging and novel scenario where the target data is out of distribution (OOD) relative to the source data and where source and target datasets have different feature spaces.

An overview of our approach is provided in Figure 2. GADT3 leverages the advantages of Test-time Training, especially the ability to adapt to new target datasets in the TTT phase without the need for labeled anomalies. Moreover, GADT3 incorporates technical innovations tailored for cross-domain GAD, including the use of domain-specific encoders, Normal Structure-preserved Attention Weighting (NSAW), and class-aware regularization. Unlike prior TTT methods that rely on auxiliary dummy tasks (e.g., rotation prediction or jigsaw solving) for encoder adaptation, our self-supervised homophily loss is directly aligned with the target anomaly detection objective. By encouraging high homophily among node embeddings in the target graph, this task-relevant signal promotes local consistency, enabling the model to adapt without labels while preserving the semantic structure essential for anomaly detection.

Our framework operates in two phases. The *training phase* combines supervised and self-supervised losses to learn from the source dataset. The *test-time training phase* only uses self-supervised learning to adapt to the target data. During training, we learn the encoder $\theta_s$, decoder $\theta_m$, and supervised predictor $\theta_{\text{pred}}$ jointly based on the source dataset. During test-time training (TTT) on the target domain, a new encoder

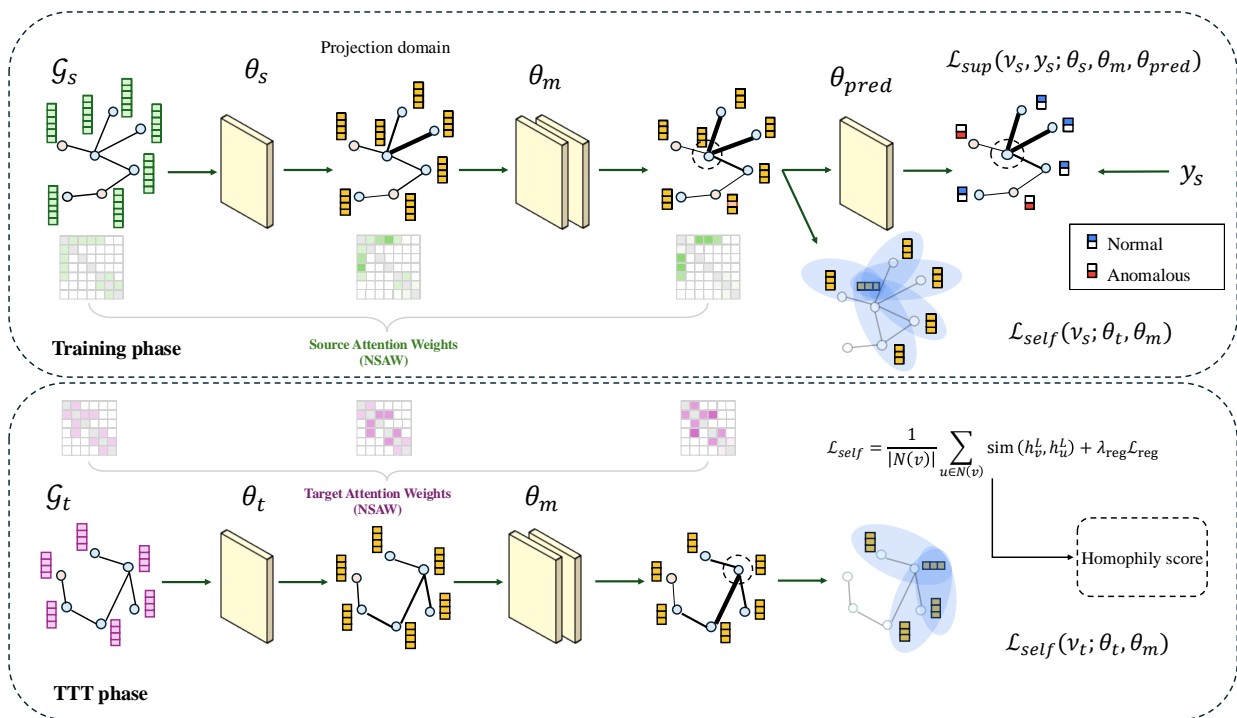

Figure 2: Overview of GADT3, our test-time training framework for node-level cross-domain anomaly detection. In the training phase (top), GADT3 learns an encoder $\theta_s$, a decoder $\theta_m$, and a prediction $\theta_{pred}$ by jointly minimizing supervised ($\mathcal{L}_{\text{sup}}$) and self-supervised homophily-based loss ($\mathcal{L}_{\text{self}}$). We propose using Normal Structure-preserved Attention Weights to defend against the adversarial influence of anomalous nodes during message passing in the GNN training. In the test-time training (TTT) phase (bottom), GADT3 learns a target encoder $\theta_t$ using only the self-supervised loss $\mathcal{L}_{\text{self}}$. The decoder $\theta_m$ is shared across phases to enable the cross-domain knowledge transfer while handling heterogeneous feature spaces. During inference, the adapted target representations produced by $\theta_m$ are used for anomaly scoring.

$\theta_t$ is trained from scratch using the unsupervised homophily-based loss $\mathcal{L}_{\text{self}}$, while $\theta_m$ is reused from the training phase. This decoupled encoder-decoder structure enables adaptation without requiring any target labels or access to source data after training. We will detail each stage of our solution together with our key contributions in the next sections.

## 4.1  Backbone GNN and Projection head

The backbone architecture of GADT3 is a Graph Neural Network (GNN) that operates on source and target graph data. Our model can apply any GNN architecture, including Message-passing Neural Networks (MPNNs). We denote the set of neighbors at node $v$ as $\mathcal{N}(v) = \{u | (u, v) \in \mathcal{E}\}$. For each layer $\ell \in \{1, 2, \ldots L\}$ of the GNN the information from neighboring nodes of $v$ is aggregated via message-passing as follows:

$$\mathbf{h}^\ell_{\mathcal{N}(v)} = M\left(\left\{\mathbf{h}^{\ell-1}_u | u \in \mathcal{N}(v)\right\}\right), \tag{1}$$

where $M(\cdot)$ is an element-wise and permutation-invariant operator. The embedding of node $v$ at the $\ell$-th layer is derived from the aggregated neighbors' embeddings and the previous embedding of $v$:

$$\mathbf{h}^\ell_v = \sigma\left(W^\ell.[\mathbf{h}^\ell_{\mathcal{N}(v)}||\mathbf{h}^{\ell-1}_v] + \mathbf{b}^\ell\right), \tag{2}$$

where $\sigma(\cdot)$ is a non-linear activation function, $||$ denotes the concatenation operator, $W^\ell \in \mathbb{R}^{p_{\ell+1} \times 2p_\ell}$, and $\mathbf{b}^\ell \in \mathbb{R}^{p_{\ell+1}}$ are trainable weight matrices and bias vectors. Representations from the last layer ($\ell = L$) can

be converted to binary class probabilities using the sigmoid function. To generate initial representations $\mathbf{h}_v^0 = P_s.\mathbf{x}_v$ for nodes with different feature sets in $D_s$ and $D_t$, GADT3 learns projection matrices (or encoders) $P_s$ and $P_t$ where, $P_s \in \mathbb{R}^{p \times p_s}$ is a trainable projection for the source data and $\mathbf{x}_v$ are node features. The same is performed for target data using the matrix $P_t$. These encoders enable projecting source and target data into a shared representation space, facilitating transfer learning.

Without loss of generality, we apply GraphSAGE (Hamilton et al., 2017) as GADT3's backbone MPNN architecture. GraphSAGE applies node sampling in the aggregation phase to increase its scalability.

## 4.2 Homophily-based Self-supervised Loss

Test-time training requires a self-supervised loss $\mathcal{L}_{self}$ applied during both training and test-time training stages. We propose the use of a previously introduced homophily-based affinity score, which is tailored for unsupervised anomaly detection (Qiao & Pang, 2024; Chen et al., 2024b).

The local affinity score measures the similarity or connection strength between a node and its neighbors. It is based on the one-class homophily phenomenon, where normal nodes tend to have a stronger affinity with their neighbors compared with anomalies. This enables these scores to be applied to unsupervised GAD. To the best of our knowledge, our work is the first to apply affinity scores for the GAD problem within the test-time training framework.

More formally, the local affinity score is computed by comparing a node's representation against those of its immediate neighbors in the graph using various similarity metrics such as cosine similarity, Euclidean distance, Jensen-Shannon divergence, and Wasserstein distance (Lin, 1991; Chen & Liu, 2020; Shen et al., 2018). By taking the average similarity with neighbors of a node $v$, we obtain a single anomaly score $s(v)$ based on how well a node is associated with its local graph structure:

$$s(v) = \frac{1}{|N(v)|} \sum_{u \in N(v)} \text{sim}(\mathbf{h}_v^L, \mathbf{h}_u^L), \tag{3}$$

where $N(v_i)$ denotes the neighbors of $v_i$, $\mathbf{h}_u^L$ represents the learned embedding for $u$, and $\text{sim}(\mathbf{a}, \mathbf{b}) = \frac{\mathbf{a}^T \mathbf{b}}{\|\mathbf{a}\|\|\mathbf{b}\|}$ describes the cosine similarity. The unsupervised task consists of maximizing the affinity score for each node.

This test-time task is scalable, as it only requires local computations, making it efficient even for large graphs. Additionally, it can be adapted to different types of graphs and node features by choosing appropriate node representation learning techniques and similarity measures. The total self-supervised loss combines the homophily-based loss with a regularization term:

$$\mathcal{L}_{\text{self}} = -\sum_{v \in \mathcal{V}} s(v) + \lambda_{\text{reg}} \mathcal{L}_{\text{reg}}, \tag{4}$$

where $\mathcal{L}_{\text{reg}} = \sum_{v \in \mathcal{V}} \left( \frac{\sum_{u \in \mathcal{N}_v^-} \text{sim}(\mathbf{h}_v, \mathbf{h}_u)}{|\mathcal{N}_v^-|} \right)$ minimizes the similarity between non-connected nodes to maintain structural distinctiveness in the embedding space, with $\mathcal{N}_v^-$ representing the set of nodes not connected to node $v$ and $\lambda$ controlling the regularization strength.

This homophily loss also serves as an implicit alignment mechanism, encouraging local consistency among neighboring node embeddings in the target graph. Since node labels are unavailable during training, the homophily loss is applied to all connected node pairs. This may include anomalous pairs, but under our homophily assumption, most edges still connect similar (normal) nodes, making the signal reliable in practice.

By leveraging the frozen source decoder $\theta_m$, the model preserves the source-learned graph patterns, achieving alignment without requiring explicit alignment terms or target labels.

It has been noted by previous work (Qiao & Pang, 2024; Chen et al., 2024b) that enforcing similarity across all connected nodes might inadvertently cause non-homophily nodes to become similar. Normal Structure-preserved Graph Truncation (NSGT) attempts to preserve normal node structure by making binary decisions

to remove edges between dissimilar nodes based on distance. However, this approach risks losing important structural information and requires setting explicit thresholds for edge removal decisions. In the next section, we introduce our enhanced message-passing approach through attention-based NSAW, a simpler and more flexible approach to suppress the adversarial effect of non-homophily edges in equation 4.

### 4.3 Normal Structure-preserved Attention Weighting (NSAW)

We propose Normal Structure-preserved Attention Weighting (NSAW) as an alternative to Normal Structure-preserved Graph Truncation (Qiao & Pang, 2024). NSAW applies a robust extension to graph attention (Veličković et al., 2017) as a more flexible way to suppress the effect of anomalous nodes on the representation of normal nodes during message-passing. Specifically, by downweighting abnormal-to-normal connections, NSAW reduces the influence of structurally inconsistent neighbors. This preserves the structure of normal regions and encourages more distinct representations for anomalies. These attention weights are learned end-to-end as part of the GNN training process.

For each layer $\ell$, we compute attention weights between connected nodes using a learnable transformation matrix $U^\ell \in \mathbb{R}^{p_\ell \times \tilde{p}}$, where $\tilde{p}$ represents the attention dimension. Given representations $H^\ell \in \mathbb{R}^{N \times p_\ell}$ at layer $\ell$, the attention scores are computed as:

$$A^\ell = \text{softmax}(\phi(H^\ell U^\ell)(\phi(H^\ell U^\ell))^\top \odot M), \tag{5}$$

where $\phi$ is ReLU activation, $M$ is the adjacency matrix, and $\odot$ is element-wise multiplication. The use of $M$ ensures that attention weights are only computed between connected nodes in the graph.

**Robust Symmetric Attention.** To defend against the adversarial influence of anomalous nodes, we enforce symmetry in the attention weights through a symmetric minimum operation:

$$\widetilde{A}^\ell_{i,j} = \min\{A^\ell_{i,j}, A^\ell_{j,i}\}, \quad \text{for all } i \neq j \tag{6}$$

NSAW's symmetric construction provides a crucial defensive mechanism through two complementary aspects. First, when a normal node $v$ connects to both normal nodes $\{v_1, v_2, \ldots, v_k\}$ and anomalous nodes $\{a_1, a_2, \ldots, a_m\}$, the attention naturally assigns high weights to normal neighbors and low weights to anomalous ones due to feature similarity. As illustrated in the Figure 3, when a normal node assigns low attention $\alpha(v \to a) \approx 0$ to an anomalous node (thin arrow), even if the anomalous node attempts to assign high attention back (thick arrow), the minimum operation ensures that the reverse influence is also minimized as $\alpha(a \to v) = \min(\alpha(a \to v), \alpha(v \to a)) \approx 0$ (dashed arrow), regardless of the original attention weight. Second, through the learnable attention parameters $U^l$ and the self-supervised loss, the architecture gradually learns to suppress the influence of anomalous patterns during

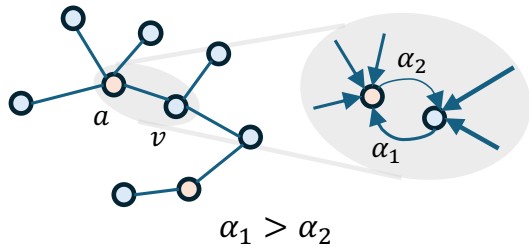

Figure 3: Symmetric attention: node $a$ gives high attention to node $v$ ($\alpha_1$), while $v$ gives lower attention to $a$ ($\alpha_2$). The final symmetric weight is $\min(\alpha_1, \alpha_2) = \alpha_2$.

training, further strengthening this defensive mechanism through the GNN's message passing structure.

**Enhanced Message Passing.** The computed attention weights modulate the message passing in the GNN architecture. For a node $v$, the attention-weighted messages from its neighbors are aggregated as follows:

$$\mathbf{h}^\ell_v = M(\{\widetilde{A}^{\ell-1}[u, v] \cdot \mathbf{h}^{\ell-1}_u \mid u \in \mathcal{N}(v)\}), \tag{7}$$

where $M(\cdot)$ combines the weighted messages using sum aggregation followed by ReLU activation.

Unlike NSGT (Qiao & Pang, 2024), which removes edges based on a binary homophily threshold, NSAW introduces a learnable and continuous mechanism to modulate edge importance. This makes NSAW both more flexible and more robust: it preserves useful structure while suppressing anomalous influences during message passing. Instead of pruning edges, NSAW learns to softly attenuate misleading connections, especially those from anomalous nodes, making it more general and data-driven for cross-domain GAD scenarios.

### 4.4 Class-aware Regularization

Class imbalance poses a major challenge in GAD. As shown in Table 5, the abnormality rate of all datasets we're using is less than 7%. Our class-aware regularization enhances the model's ability to handle class imbalance during the source training phase by applying stronger regularization to the minority class (i.e., anomalous nodes). When minimizing similarity between non-connected nodes, it assigns higher weights to anomalous nodes, ensuring they maintain their distinctive patterns rather than being overshadowed by the majority class representations. Here's the mathematical formulation:

$$\mathcal{L}_s = \sum_{i \in \mathcal{V}} \left( \frac{\sum_{j \in \mathcal{N}_i^-} w_j \cdot \mathrm{sim}(\mathbf{h}_i, \mathbf{h}_j)}{|\mathcal{N}_i^-|} \right), \tag{8}$$

where $\mathcal{N}_i^-$ represents the set of nodes not connected to node $i$, $\mathrm{sim}(h_i, h_j)$ is the cosine similarity between node embeddings, and $w_j$ is the class-based weight defined as:

$$w_j = \begin{cases} \alpha & \text{if node } j \text{ is anomalous} \\ 1 & \text{otherwise} \end{cases} \tag{9}$$

where $\alpha > 1$ is the weighting factor for anomalous nodes. We set $\alpha$ inversely proportional to the class ratio This regularization can only be applied during source training, where we have access to node labels, while target training relies solely on the homophily-based loss due to its unsupervised nature.

### 4.5 Test-Time Training

We can now describe our framework for anomaly detection with domain adaptation (GADT3), summarized in Figure 2. Following the test-time training paradigm, GADT3 is based on two tasks, the *main (supervised) task* and the *auxiliary self-supervised learning (SSL) task*. In the training phase, GADT3 is trained based on both tasks using a joint loss minimized based on the source dataset. Subsequently, the test-time training phase leverages only the SSL task using the target dataset, which is assumed to be unlabeled.

**Training phase.** During training, GADT3 applies the source graph $\mathcal{G}_s$ to learn an encoder $\theta_s$, a decoder $\theta_m$, and a predictor $\theta_{\mathrm{pred}}$ by minimizing a loss function $\mathcal{L}_{\mathrm{train}}$ that is a combination of a supervised loss $\mathcal{L}_{\mathrm{sup}}$, for which we apply cross-entropy, and a self-supervised loss $\mathcal{L}_{\mathrm{self}}$, for which we apply the homophily-based affinity scores (see Sec. 4.2):

$$\mathcal{L}_{\mathrm{sup}} = \sum_{v \in \mathcal{V}_s} y_v \log \hat{y}_v + \lambda_s \mathcal{L}_s, \tag{10}$$

$$\mathcal{L}_{\mathrm{self}} = -\sum_{v \in \mathcal{V}} \left( \frac{1}{|N(v)|} \sum_{u \in N(v)} \mathrm{sim}(\mathbf{h}_v^L, \mathbf{h}_u^L) \right) + \lambda_{\mathrm{reg}} \mathcal{L}_{\mathrm{reg}}, \tag{11}$$

$$\mathcal{L}_{\mathrm{train}} = \sum_{v \in \mathcal{V}_s} \left( \mathcal{L}_{\mathrm{sup}}(v, y; \theta_m, \theta_s, \theta_{\mathrm{pred}}) + \lambda \mathcal{L}_{\mathrm{self}}(v; \theta_m, \theta_s) \right), \tag{12}$$

where $\hat{y}$ is the predicted label (normal or anomaly) and $\lambda$, $\lambda_s$, and $\lambda_{\mathrm{reg}}$ are weight hyperparameters.

We describe the source encoder $\theta_s$ (projection) and decoder $\theta_m$ (GNN) in Section 4.1. The predictor $\theta_{\mathrm{pred}}$ is a multi-layer perceptron that maps node embeddings to label predictions.

**Test-time training phase.** During the test-time training (TTT) phase, we apply the target graph $\mathcal{G}_t$ to optimize a target encoder $\theta_t$ by minimizing the TTT loss $\mathcal{L}_{\mathrm{TTT}}$:

$$\mathcal{L}_{\mathrm{TTT}} = \sum_{v \in \mathcal{V}_t} \mathcal{L}_{\mathrm{self}}(v; \theta_t, \theta_m), \tag{13}$$

where $\mathcal{L}_{\text{self}}$ is from Eqn. 11 and the decoder $\theta_m$ is reused from the training phase to embed target nodes.

**Inference:** Anomaly detection in the target domain is performed using the adapted encoder $\theta_t$ optimized during test-time training (TTT) based solely on the target graph. The decoder $\theta_m$, trained on the source domain, is reused without modification. The encoder $\theta_t$ maps target node features into a shared embedding space, and $\theta_m$ further refines these representations using the target graph. We do not use the prediction head $\theta_{\text{pred}}$ at inference time. Instead, we compute homophily scores in the embedding space generated by $\theta_m$, rank the nodes based on these scores, and directly evaluate performance using AUROC and AUPRC.

Although $\theta_{\text{pred}}$ is not applied during inference, we observe the embeddings learned by the adapted target encoder $\theta_t$ remain well-aligned with the source decision boundary learned during training. This is shown in Figure 4, where we show that source and target embeddings share similar separation patterns, indicating that the shared representation space preserves discriminative structure even without explicit prediction. While $\theta_{\text{pred}}$ is trained using source labels, it can technically be reused during inference. However, GADT3 relies on homophily-based node ranking. See Appendix A.6 for an alternative approach.

**Stopping criteria.** We propose an adaptive early-stopping strategy based on distance ratios between source and target domains. Our method computes the distance between target features and the average embeddings of two distinct source classes—normal and anomalous. The ratio of maximum to minimum distance serves as the stopping criterion, where a higher ratio indicates better domain adaptation as it reflects stronger alignment between target samples and their corresponding source class features. The training stops when no improvement in this ratio is observed for a predefined number of epochs. Our ratio-based approach measures relative alignment between target samples and source class distributions, rather than just overall domain similarity. See Appendix A.1 for details.

## 4.6 Theoretical Guarantee for TTT in GADT3

We propose a formal guarantee for the case of homogeneous feature spaces across domains. Under the Test-Time Training (TTT) update in GADT3, the separation margin between normal and anomalous nodes in the target graph monotonically increases. This establishes that our homophily-based self-supervised objective provably improves anomaly separability during adaptation.

We begin by introducing the notation used in the proof. Let $\theta_s^*$ denote the source encoder learned during the supervised pre-training phase, and let $\theta_t^{(k)}$ be the target encoder parameter learned during step $k$ of TTT. Recall that the decoder $\theta_m$ is kept fixed throughout TTT. For a target node $v$, we denote by $s_t(v; \theta_t)$ its homophily score computed with encoder $\theta_t$. The *anomaly separation margin* at step $k$ is defined as the difference between expected values of homophily scores for normal and anomalous nodes in the graph:

$$\Delta_t^{(k)} = \mathbb{E}_{v_N \in \mathcal{N}}[s_t(v_N; \theta_t^{(k)})] - \mathbb{E}_{v_A \in \mathcal{A}}[s_t(v_A; \theta_t^{(k)})],$$

where $\mathcal{N}$ and $\mathcal{A}$ denote the sets of normal and anomalous nodes in the target graph, respectively. We further define the *class-conditional gradient* for class $C \in \{\mathcal{N}, \mathcal{A}\}$ as:

$$g_C(\theta) = \nabla_\theta \mathbb{E}_{v \in C}[s_t(v; \theta)].$$

In particular, we denote $g_N^{-1} = g_\mathcal{N}(\theta_s^*)$ and $g_A^{-1} = g_\mathcal{A}(\theta_s^*)$ as the class-conditional gradients at $\theta_s^\star$. Moreover, we denote $g_N^{(k)} = g_\mathcal{N}(\theta_t^{(k)})$ and $g_A^{(k)} = g_\mathcal{A}(\theta_t^{(k)})$ the target gradients at step $k$ of TTT.

We assume that $\theta_s$ and $\theta_t$ share the same feature dimension size ($p_s = p_t$). Moreover, for the optimal source encoder $\theta_s^*$, we have the following assumptions:

**Assumption 1.** *(Gradient dominance at source optimum)* $\|g_N^{-1}\| > \|g_A^{-1}\|$, *i.e., the magnitude of the gradient induced by normal nodes is strictly greater than that induced by anomalous nodes.*

**Assumption 2.** *(Lipschitz continuity and boundedness)* *The class-conditional gradients are $L$-Lipschitz continuous and uniformly bounded: $\|g_C(\theta_1) - g_C(\theta_2)\| \leq L\|\theta_1 - \theta_2\|$ and $\|g_C(\theta)\| \leq G$ for all $\theta_1, \theta_2, \theta$ and for $C \in \{\mathcal{N}, \mathcal{A}\}$.*

**Assumption 3.** *(Initialization proximity)* *The target encoder initialization $\theta_t^{(0)}$ is sufficiently close to the optimal source encoder $\theta_s^*$.*

The above assumptions are reasonable given that the source encoder $\theta_s^*$ is trained with labeled data, which naturally yields stronger gradient signals from normal nodes in the homophily-based objective. It does, however, require $\theta_s$ and $\theta_t$ to share the same feature dimension size ($p_s = p_t$). Moreover, at the source optimum, normal nodes are more homophilic and consistent, so their gradients align in direction and yield a larger $\|g_N^{-1}\|$ than scattered anomalous gradients.

**Proposition 1** (Monotonic Margin Increase)**.** *Under the above assumptions, for all $k \geq 0$, starting from an initial parameter $\theta_t^{(0)}$ sufficiently close to the optimal $\theta_t^*$, the TTT update satisfies:*

$$\Delta_t^{(k+1)} > \Delta_t^{(k)}.$$

That is, each step of test-time training strictly increases the anomaly separation margin between normal and anomalous nodes.

## 5 Experiments

We evaluate the proposed approach (GADT3) against multiple baselines across six graph datasets, including methods from both cross-domain adaptation and anomaly detection. Our analysis is complemented with ablation studies on key components of the model, such as source training, NSAW, and class-aware regularization. To assess scalability, we further test GADT3 on three large-scale graphs. Additionally, we investigate the model's sensitivity to the homophily assumption by synthetically varying homophily levels to identify the threshold at which performance begins to degrade.

### 5.1 Experimental Setup

**Datasets.** We apply six datasets from diverse domains, such as online shopping reviews, including Amazon (AMZ) (McAuley & Leskovec, 2013), YelpChi (Rayana & Akoglu, 2015), YelpHotel (HTL), and YelpRes (RES) (Ding et al., 2021), and social networks, including Reddit (RDT) (Kumar et al., 2018) and Facebook (FB)(Leskovec & Mcauley, 2012). For our cross-domain analysis, we examine scenarios where feature spaces are homogeneous (same features) and heterogeneous (different features) across domains. To demonstrate the scalability of our model, we additionally report results on three large-scale graphs: Amazon-all (AMZ-all), YelpChi-all (YC-all) (Hamilton et al., 2017), and T-Finance (TF) (Tang et al., 2022) in Table 3. A detailed description of each dataset is provided in Appendix A.2.

**Baselines and Evaluation Criteria.** We evaluate our method against state-of-the-art baselines both for cross-domain domain adaptation and (single-graph) unsupervised anomaly detection. Cross-domain baselines include GRADE (Wu et al., 2023), AdaGCN (Dai et al., 2022), UDA-GCN (Wu et al., 2020), and ACT (Wang et al., 2023b). Test-time training baselines include TENT (Wang et al., 2020), GraphCL (You et al., 2020), and GTrans (Jin et al., 2022). As GAD baselines, we consider self-supervised learning-based methods (CoLA (Liu et al., 2021a) and SL-GAD (Zheng et al., 2021)), reconstruction-based methods (DOMINANT (Ding et al., 2019) and ComGA (Luo et al., 2022)), and a local affinity-based method (TAM (Qiao & Pang, 2024)).

While GADT3 assumes that normal nodes are more homophilic than anomalies, most baselines (e.g., DOMINANT, CoLA, SL-GAD, GTrans) are homophily-agnostic. Yet, GADT3 outperforms them, indicating that homophily-based self-supervision provides a useful inductive bias for cross-domain anomaly detection.

We utilize two metrics to evaluate anomaly detection models: the Area Under the Receiver Operating Characteristic Curve (AUROC) and the Area Under the Precision-Recall Curve (AUPRC). Higher values of AUROC and AUPRC indicate superior model performance. We report average results for 10 repetitions.

**Implementation details.** We developed GADT3 using Python 3.11.9 and PyTorch 2.1.0. All the experiments were conducted on an NVIDIA A40 GPU with 48GB. The core model is a 2-layer GraphSAGE GNN with weight parameters optimized via the Adam optimizer (Kingma & Ba, 2014) with a dropout rate of 0.7. The source model underwent training for 100 epochs with early stopping employed to determine the target epochs. Both the source and target models were trained with a learning rate of 0.001. We set $\lambda = 0.001$, $\lambda_{\text{reg}} = 0.1$, $\lambda_s = 0.001$, $p = 40$, and $\alpha = 20$. Baselines were trained using the hyperparameters recommended in their respective papers. For hyperparameter sensitivity analysis, please refer to Appendices A.3 and A.8.

Table 1: Cross-domain (Source → Target) node-level graph anomaly detection performance of GADT3 compared to state-of-the-art baselines. We have shown the results for domain pairs with both homogeneous and heterogeneous feature spaces. Results show GADT3 significantly outperforms baselines in most settings.

| Metric (%) | Type | Method | Heterogeneous Features | | | | | | Homogeneous Features | | Avg. |
|---|---|---|---|---|---|---|---|---|---|---|---|
| | | | AMZ→RDT | AMZ→FB | RDT→AMZ | RDT→FB | FB→AMZ | FB→RDT | HTL→RES | RES→HTL | |
| **AUROC** | CDA | GRADE | $61.80_{\pm0.8}$ | $84.96_{\pm0.7}$ | $70.38_{\pm1.2}$ | $85.53_{\pm0.6}$ | $72.06_{\pm0.9}$ | $61.27_{\pm1.0}$ | $75.51_{\pm0.6}$ | $73.80_{\pm0.5}$ | $73.16_{\pm0.4}$ |
| | | AdaGCN | $58.86_{\pm0.7}$ | $72.35_{\pm0.8}$ | $70.43_{\pm0.6}$ | $75.89_{\pm1.1}$ | $64.86_{\pm1.3}$ | $61.62_{\pm0.9}$ | $79.50_{\pm0.7}$ | $81.18_{\pm0.5}$ | $70.59_{\pm0.6}$ |
| | | UDA-GCN | $60.37_{\pm1.0}$ | $80.80_{\pm0.9}$ | $67.36_{\pm1.2}$ | $84.98_{\pm0.6}$ | $67.55_{\pm1.0}$ | $\mathbf{61.95}_{\pm0.8}$ | $85.36_{\pm0.9}$ | $78.24_{\pm0.6}$ | $73.33_{\pm0.7}$ |
| | | ACT | $59.24_{\pm1.3}$ | $81.97_{\pm0.5}$ | $71.16_{\pm0.6}$ | $75.74_{\pm1.1}$ | $71.98_{\pm1.0}$ | $60.52_{\pm1.2}$ | $89.20_{\pm0.4}$ | $80.40_{\pm0.8}$ | $73.78_{\pm0.6}$ |
| | TTT | TENT | $59.01_{\pm0.9}$ | $76.15_{\pm0.8}$ | $71.51_{\pm0.5}$ | $83.43_{\pm0.7}$ | $66.88_{\pm1.2}$ | $58.20_{\pm0.9}$ | $75.11_{\pm1.0}$ | $79.27_{\pm0.5}$ | $71.20_{\pm0.5}$ |
| | | GraphCL | $60.37_{\pm0.7}$ | $88.46_{\pm0.6}$ | $68.71_{\pm0.8}$ | $88.81_{\pm0.7}$ | $61.91_{\pm1.0}$ | $61.64_{\pm0.9}$ | $79.05_{\pm0.5}$ | $74.20_{\pm0.6}$ | $72.89_{\pm0.7}$ |
| | | GTrans | $60.64_{\pm0.6}$ | $89.35_{\pm0.5}$ | $73.93_{\pm0.9}$ | $90.18_{\pm0.4}$ | $68.03_{\pm0.7}$ | $61.90_{\pm0.8}$ | $79.86_{\pm0.7}$ | $83.49_{\pm0.5}$ | $75.92_{\pm0.6}$ |
| | Ours | **GADT3** | $\mathbf{61.95}_{\pm0.5}$ | $\mathbf{91.03}_{\pm0.2}$ | $\mathbf{79.87}_{\pm0.3}$ | $\mathbf{94.84}_{\pm0.3}$ | $\mathbf{82.57}_{\pm0.3}$ | $61.93_{\pm0.7}$ | $\mathbf{94.14}_{\pm0.1}$ | $\mathbf{90.71}_{\pm0.1}$ | $\mathbf{82.13}_{\pm0.3}$ |
| **AUPRC** | CDA | GRADE | $4.92_{\pm0.2}$ | $18.44_{\pm0.3}$ | $11.87_{\pm0.5}$ | $16.10_{\pm0.4}$ | $13.00_{\pm0.3}$ | $4.34_{\pm0.2}$ | $18.81_{\pm0.6}$ | $22.53_{\pm0.4}$ | $13.75_{\pm0.3}$ |
| | | AdaGCN | $4.51_{\pm0.3}$ | $9.04_{\pm0.2}$ | $12.15_{\pm0.4}$ | $5.72_{\pm0.5}$ | $10.23_{\pm0.4}$ | $4.64_{\pm0.3}$ | $22.15_{\pm0.4}$ | $35.15_{\pm0.5}$ | $12.95_{\pm0.3}$ |
| | | UDA-GCN | $4.65_{\pm0.2}$ | $6.73_{\pm0.3}$ | $10.85_{\pm0.3}$ | $22.04_{\pm0.6}$ | $10.89_{\pm0.4}$ | $5.55_{\pm0.3}$ | $26.02_{\pm0.5}$ | $27.31_{\pm0.4}$ | $14.26_{\pm0.3}$ |
| | | ACT | $4.22_{\pm0.4}$ | $8.16_{\pm0.3}$ | $13.20_{\pm0.5}$ | $26.52_{\pm0.4}$ | $12.26_{\pm0.3}$ | $4.88_{\pm0.2}$ | $33.00_{\pm0.3}$ | $28.70_{\pm0.5}$ | $16.37_{\pm0.4}$ |
| | TTT | TENT | $4.60_{\pm0.3}$ | $11.69_{\pm0.4}$ | $12.49_{\pm0.3}$ | $18.81_{\pm0.5}$ | $12.32_{\pm0.4}$ | $4.79_{\pm0.3}$ | $13.12_{\pm0.3}$ | $28.31_{\pm0.4}$ | $13.27_{\pm0.4}$ |
| | | GraphCL | $4.69_{\pm0.2}$ | $21.09_{\pm0.3}$ | $14.36_{\pm0.4}$ | $30.86_{\pm0.3}$ | $11.12_{\pm0.5}$ | $5.44_{\pm0.3}$ | $21.27_{\pm0.4}$ | $25.55_{\pm0.3}$ | $16.80_{\pm0.3}$ |
| | | GTrans | $4.78_{\pm0.3}$ | $25.95_{\pm0.4}$ | $17.02_{\pm0.5}$ | $29.11_{\pm0.2}$ | $11.91_{\pm0.4}$ | $5.15_{\pm0.3}$ | $21.03_{\pm0.4}$ | $36.20_{\pm0.3}$ | $18.89_{\pm0.4}$ |
| | Ours | **GADT3** | $\mathbf{5.19}_{\pm0.1}$ | $\mathbf{27.06}_{\pm0.1}$ | $\mathbf{19.10}_{\pm0.4}$ | $\mathbf{34.76}_{\pm0.3}$ | $\mathbf{25.73}_{\pm0.4}$ | $\mathbf{6.12}_{\pm0.2}$ | $\mathbf{39.75}_{\pm0.1}$ | $\mathbf{42.90}_{\pm0.3}$ | $\mathbf{25.08}_{\pm0.3}$ |

## 5.2 Cross-domain Graph Anomaly Detection

We present the results comparing our approach against various domain adaptation baselines in Table 1. We use different datasets as source and target domains. We incorporated a projection head into all baseline models to address mismatches in feature sets between source and target domains. This ensures a fair comparison by allowing each method to handle discrepancies in input feature spaces. In homogeneous settings, where source and target graphs share the same feature space, no projection head was used.

Our proposed method (GADT3) outperformed the baselines in most scenarios. GADT3 achieved the highest AUROC and AUPRC scores in 5 out of 6 tasks. For instance, in terms of AUROC, in the Reddit→Amazon task, GADT3 surpassed the next best method (GTrans) by 5.9%. In the Reddit→Facebook task, GADT3 outperformed the closest competitor (GTrans) by 4.7%. The only task where GADT3 did not lead in terms of AUROC and AUPRC was Facebook→Reddit, where it performed competitively but slightly below UDA-GCN. On average, our method outperformed the second-best baseline by at least 6.2 points in both AUROC and AUPRC. In Appendix A.5, we provide further insights into the performance of GADT3. We have found that high target performance is often correlated with high source homophily. This can be explained by our use of a homophily-based self-supervised loss for test-time training.

## 5.3 Representation Embedding

The effectiveness of our TTT approach is demonstrated in Figure 4, which presents the two-dimensional T-SNE embeddings of the GNN outputs for both source and target models. On the left side, we observe a clear separation between normal and anomalous samples in the source domain (Amazon), indicating that the source model has effectively learned to distinguish between classes. The right figure shows the transferability of our approach to the target domain (Facebook). Despite the inherent differences between source and target datasets, the adapted embeddings in the target domain preserve meaningful structure, demonstrating that our TTT strategy successfully refines the target representations using the pre-trained source model. We provide additional visualizations of source and target embeddings for other datasets in Appendix A.12.

## 5.4 Early-Stopping Mechanism

This section evaluates our early-stopping strategy across different domain adaptation scenarios. Figure 5 illustrates how the distance ratio evolves together with the Area Under the Precision-Recall Curve (AUPRC) during training. We marked the specific epoch when the model was selected. The results show that the ratio effectively captures the convergence of domain adaptation without the need for labeled validation data.

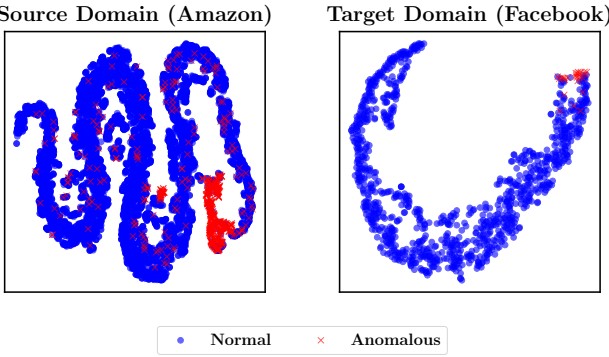

Figure 4: T-SNE embeddings of node representations for source domain (Amazon) on the left and target domain (Facebook) on the right. The source model clearly separates normal/anomalous nodes. After adaptation, the pre-trained model maintains this separation in the target domain, indicating successful transfer.

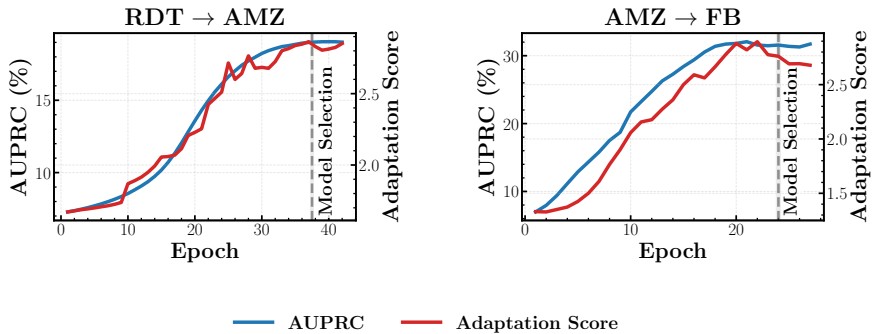

Figure 5: Relationship between target anomaly detection accuracy (AUPRC) and our ratio-based adaptation score for Reddit→Amazon and Amazon→Facebook. We use the adaptation score as a label-free early stopping criterion, which is triggered at epochs 37 (left) and 24 (right).

## 6 Ablation Studies

We perform four ablation studies to evaluate different aspects of GADT3. The results support the design of our model and demonstrate the relevance of each of its components.

**GAD Results (Source-free)**. We evaluate a single-domain version of GADT3 against six state-of-the-art GAD baselines using four datasets. Our goal is to isolate the impact of GADT3's self-supervised learning from its cross-domain adaptation. The results are shown in Table 2. We follow the same protocol and use the same datasets as Qiao & Pang (2024), so we report their results for the baselines. GADT3 outperforms baselines even in the source-free setting, as the NSAW edge weighting effectively leverages structural context to enhance anomaly separation without relying on source data.

**NSAW as a learnable and flexible edge weighting approach.** We proposed NSAW as a learnable edge-weighting mechanism to suppress the adverse effect of non-homophily edges in anomaly detection. Table 2 compares our method against TAM (Qiao & Pang, 2024) (using NSGT). The results show that GADT3 outperforms TAM using most datasets. This is evidence that NSAW is a more flexible alternative to NSGT by allowing GADT3 to learn continuous attention weights. To validate its effectiveness, we compare GADT3 with and without NSAW under our cross-domain tasks. Introducing NSAW improves AUROC from 90.83% to 91.04% on AMZ→FB and from 81.63% to 82.57% on FB→AMZ. These gains confirm that learnable, structure-preserved edge weighting is more effective than static edge removal strategies.

Table 2: GADT3, trained on target only, outperforms specialized baselines on most datasets.

| Metric (%) | Method | AMZ | RDT | FB | YC | Avg. |
|---|---|---|---|---|---|---|
| | CoLA | 58.98 | 60.28 | 84.34 | 46.36 | 62.49 |
| | SL-GAD | 59.37 | 56.77 | 79.36 | 33.12 | 57.16 |
| | DOMINANT | 59.96 | 55.55 | 56.77 | 41.33 | 53.40 |
| AUROC | ComGA | 58.95 | 54.53 | 60.55 | 43.91 | 54.48 |
| | TAM | 70.64 | 60.23 | 91.44 | **56.43** | 69.69 |
| | **GADT3** | **84.76** | **61.54** | **91.74** | 48.67 | **71.68** |
| | CoLA | 6.77 | 4.49 | 21.06 | 4.48 | 9.20 |
| | SL-GAD | 6.34 | 4.06 | 13.16 | 3.50 | 6.76 |
| | DOMINANT | 14.24 | 3.56 | 3.14 | 3.95 | 6.22 |
| AUPRC | ComGA | 11.53 | 3.74 | 3.54 | 4.23 | 5.76 |
| | TAM | 26.34 | 4.46 | 22.33 | **7.78** | 15.23 |
| | **GADT3** | **27.53** | **6.01** | **26.13** | 6.14 | **16.45** |

Table 3: AUROC and AUPRC (%) on large-scale graphs. **Bold** indicates best.

| Metric (%) | Method | AMZ-all | YC-all | TF |
|---|---|---|---|---|
| | CoLA | 26.14 | 48.01 | 48.29 |
| | SL-GAD | 27.28 | 55.51 | 46.48 |
| | DOMINANT | 69.37 | 53.90 | 53.80 |
| AUROC | ComGA | 71.54 | 53.52 | 55.42 |
| | TAM | 84.76 | **58.18** | 61.75 |
| | **GADT3** | **86.23** | 55.31 | **62.91** |
| | CoLA | 5.16 | 13.61 | 4.10 |
| | SL-GAD | 4.44 | 17.11 | 3.86 |
| | DOMINANT | 10.15 | 16.38 | 4.74 |
| AUPRC | ComGA | 18.54 | 16.58 | 4.81 |
| | TAM | 43.46 | **18.86** | 5.47 |
| | **GADT3** | **44.98** | 17.77 | **6.23** |

**Performance on Large-Scale Graphs.** We evaluate GADT3 on three large-scale datasets and compare it against strong GAD baselines in Table 3. GADT3 outperforms all baselines on Amazon-all and T-Finance, and performs competitively on YelpChi-all.

**Robustness Under Low Homophily Settings.** Our self-supervised loss for anomaly detection is based on homophily, *i.e.*, the tendency of similar nodes to connect. To assess the generalization of GADT3 in settings where this assumption weakens and beyond homophilic graphs, we synthetically rewired a portion of edges to reduce homophily in the AMZ and RDT datasets. As shown in Table 4, GADT3 remained robust when the target domain's average homophily score was $\geq 0.5$, with only gradual degradation in performance below that threshold. These results show the robustness of our method while acknowledging its reliance on moderate levels of homophily.

Table 4: AUROC (%) under decreasing target-domain homophily.

| Homophily | AMZ→RDT | RDT→AMZ |
|---|---|---|
| 0.9 | 62.3 | 81.3 |
| 0.7 | 62.7 | 79.6 |
| 0.5 | 60.9 | 79.1 |
| 0.3 | 58.3 | 63.1 |
| 0.1 | 58.8 | 63.7 |

## 7 Conclusion

In this paper, we have introduced GADT3, a novel TTT framework for GAD when the testing data is out-of-distribution and from heterogeneous application domains. Specifically, our approach addresses scenarios where the distributions and feature spaces differ between the source and target datasets using dataset-specific encoders. GADT3 combines the advantages of TTT with multiple innovations focused on graph anomaly detection. For instance, by leveraging the distinctive distribution pattern of node homophily between normal and anomalous nodes, we have presented a tailored SSL task along with a robust attention-based edge-weighting mechanism (NSAW) to enhance the generalization of the learned representations. Moreover, GADT3 performs model selection using a distance ratio-based early-stopping criterion. Our experiments have illustrated that GADT3 outperforms state-of-the-art graph domain adaptation baselines.

**Limitations:** One current limitation of our approach is its reliance on the assumption that normal nodes exhibit higher homophily compared to anomalous ones. While this pattern holds across many real-world datasets, relaxing the homophily assumption remains an important direction for future work to further enhance generalization to highly heterophilic graphs. Moreover, we will study theoretical guarantees for cross-domain GAD based on the similarity between source and target datasets in future work.

## 8   Acknowledgments

We gratefully acknowledge the support of the US Department of Transportation (USDOT) Tier-1 University Transportation Center (UTC) Transportation Cybersecurity Center for Advanced Research and Education (CYBER-CARE) (Grant No. 69A3552348332) and the Rice University Ken Kennedy Institute.

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
