# A    Appendix

## A.1    Early stopping criterion

We propose an adaptive early-stopping strategy that monitors distribution shifts between source and target domains using a distance-based metric (**Dis**). Let $f_t \in \mathbb{R}^{n_t \times p}$ denote the target embedding, and let $\mu_s^n \in \mathbb{R}^p$ and $\mu_s^a \in \mathbb{R}^p$ denote the pre-computed average normal and attack source embeddings, respectively, where $n_t$ is the number of target samples and $p$ is the feature dimension.

The distance scores are computed as:

$$\text{Dis}_n^i = \|f_t[i] - \mu_s^n\|_2 \tag{14}$$

$$\text{Dis}_a^i = \|f_t[i] - \mu_s^a\|_2 \tag{15}$$

The early stopping score is then defined as:

$$\text{score} = \frac{1}{|\mathcal{V}|} \sum_{i \in \mathcal{V}} \frac{\max(\text{Dis}_n^i, \text{Dis}_a^i)}{\min(\text{Dis}_n^i, \text{Dis}_a^i)} \tag{16}$$

For target attack samples, $\text{Dis}_n$ would be larger than $\text{Dis}_a$, making the score $\text{Dis}_n/\text{Dis}_a$, which increases as samples align better with attack source features. Conversely, for target normal samples, the score becomes $\text{Dis}_a/\text{Dis}_n$, increasing as samples align with normal source features. The training stops at epoch $T$ if:

$$\text{score}(t) \le \text{score}^*(t - r) \quad \forall t \in [T - r + 1, T] \tag{17}$$

where $\text{score}^*(t)$ is the best score up to epoch $t$ and $r$ is the patience parameter. A higher score indicates better domain adaptation as it represents a stronger alignment with the correct source class features.

## A.2    Data description

**Amazon** (McAuley & Leskovec, 2013): The Amazon dataset consists of product reviews from the Musical Instruments category. Users with more than 80% helpful votes are labeled as benign entities, while those with less than 20% helpful votes are considered fraudulent entities. For each user (represented as a node in the graph), 25 handcrafted features are used as raw node features. The graph structure is defined by the U-P-U (User-Product-User) relation, which connects users who have reviewed at least one common product.

**Reddit** (Kumar et al., 2018): The Reddit dataset consists of forum posts from the Reddit platform, focusing on user behavior and content. In this dataset, users who have been banned from the platform are labeled as anomalies. Each post's textual content has been vectorized to serve as one of the 64 attributes for the corresponding user or post.

**Facebook** (Leskovec & Mcauley, 2012): The dataset represents a social network derived from the Facebook platform. Users establish connections with other users, forming a network of relationships. Fraudulent users are assumed to be anomalies of the network. Each node represents one user with 576 node attributes.

**YelpChi** (Rayana & Akoglu, 2015): The dataset comprises hotel and restaurant reviews, categorized as either filtered (spam) or recommended (legitimate) by Yelp. We utilize 32 handcrafted features as raw node features for each review in the Yelp dataset. In the graph, the reviews serve as nodes. The graph's structure is defined by the R-U-R (Review-User-Review) relation, which connects reviews posted by the same user.

**YelpHotel** (Ding et al., 2021): Graph dataset based on hotel reviews from Yelp, where users and hotels are nodes connected by review edges. Each review contains ratings, text, and detailed metadata about both hotels and reviewers.

**YelpRes** (Ding et al., 2021): Graph dataset containing restaurant reviews from Yelp, where users and restaurants are nodes connected by review edges. Each review has ratings, text, and metadata about both users and restaurants

Table 5 presents a summary of the dataset statistics.

Table 5: Statistics of the benchmark datasets used in our experiments.

| Dataset | # nodes | # edges | # features | Abnormal (%) |
|---|---|---|---|---|
| Amazon (AMZ) | 10,244 | 175,608 | 25 | 6.66 |
| Reddit (RDT) | 10,984 | 168,016 | 64 | 3.33 |
| Facebook (FB) | 1,081 | 55,104 | 576 | 2.49 |
| YelpChi (YC) | 24,741 | 49,315 | 32 | 4.91 |
| YelpHotel (HTL) | 4,322 | 101,800 | 8,000 | 5.78 |
| YelpRes (RES) | 5,012 | 355,144 | 8,000 | 4.99 |
| Amazon-all (AMZ-all) | 11,944 | 4,398,392 | 25 | 6.87 |
| YelpChi-all (YC-all) | 45,941 | 3,846,979 | 32 | 14.52 |
| T-Finance (TF) | 39,357 | 21,222,543 | 10 | 4.58 |

Table 6: Ablation study of GADT3 with and without class-aware regularization. Results show AUROC (%) and AUPRC (%) across domain pairs. **Bold** indicates better performance.

| Datasets | AUROC (%) | | AUPRC (%) | |
|---|---|---|---|---|
| | w/ | w/o | w/ | w/o |
| AMZ→RDT | **61.95** | 61.76 | **5.19** | 4.93 |
| AMZ→FB | **91.03** | 90.83 | **27.06** | 27.01 |
| RDT→AMZ | 79.87 | **82.10** | **19.10** | 18.19 |
| RDT→FB | **94.84** | 90.73 | **34.76** | 32.71 |
| FB→AMZ | **82.57** | 81.98 | **25.73** | 24.98 |
| FB→RDT | 61.93 | **61.98** | **6.12** | 4.87 |

## A.3 Hyperparameter tuning

We tune the main hyperparameters of GADT3, including the self-supervision weight $\lambda$, the regularization strength $\lambda_{\text{reg}}$, and the class-aware weighting factor $\alpha$. A grid search is performed using a small labeled validation set from the source (not the target) domain. Specifically, we search $\lambda \in 0.0001, 0.0005, 0.001, 0.005, 0.01$ on a logarithmic scale, and fix $\lambda_{\text{reg}} = 0.1$ and $\lambda_s = 0.001$ based on prior ablation results. We set $\alpha$ inversely proportional to the anomaly ratio as $\alpha = 1/r$, where $r$ is the proportion of anomalous nodes. As shown in Figure 6, $\lambda = 0.001$ consistently achieves the best or near-best performance across source domains, making it a robust choice for cross-domain transfer.

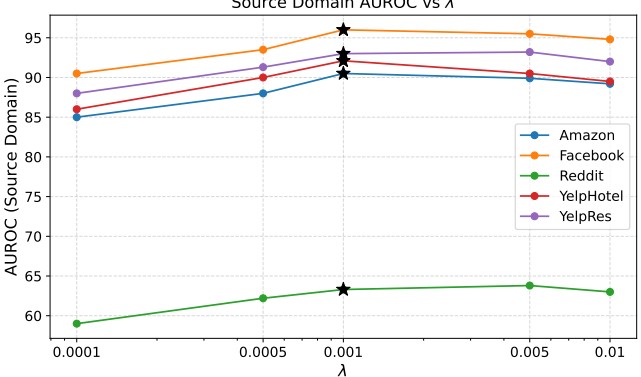

Figure 6: AUROC on source domains across varying $\lambda$ values. Most datasets peak at $\lambda = 0.001$, indicating it as a robust default for adaptation.

## A.4 Class-aware regularization.

In Table 6, we compare the performance of GADT3 with and without the class-aware source regularization approach described in 4.4. The results show that the regularization helps increase the sensitivity of the source model to anomalous nodes. Regularization improves the accuracy of the model in most settings. In particular, the results show that regularization improves the performance in terms of AUPRC, which is better at capturing class imbalance than AUROC.

## A.5 Transfer performance with respect to source-target homophily

Our analysis reveals that the direction of transfer plays a crucial role in cross-domain anomaly detection performance. Transferring from a higher homophily domain to a lower homophily domain generally results

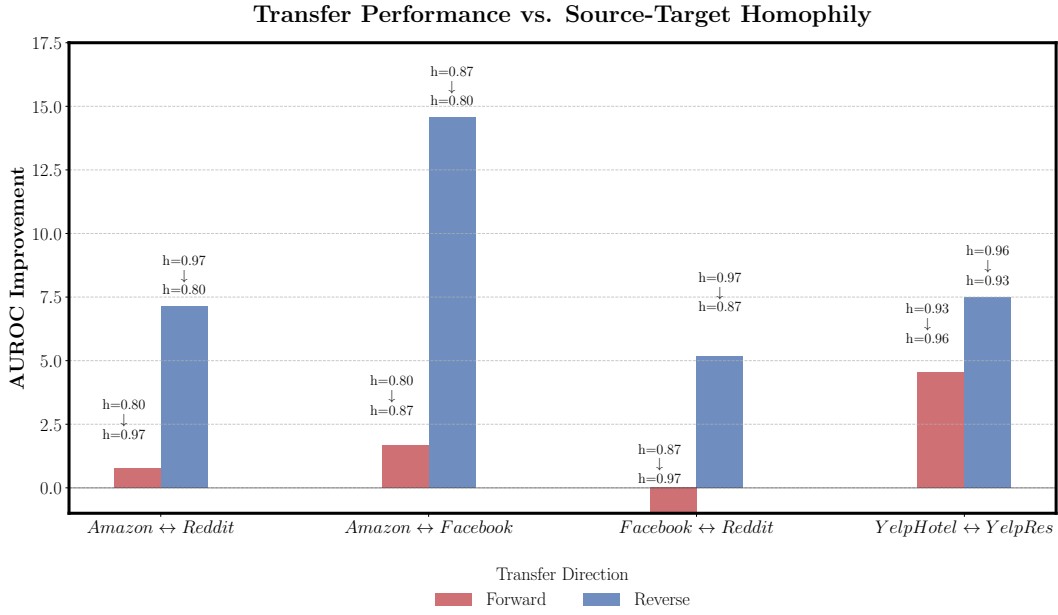

Figure 7: Domains with higher homophily (h) generally serve as better source domains when transferring to domains with lower homophily.

Table 7: Cross-domain node-level anomaly detection results for GADT3 using frozen prediction layer and original GADT3.

| Metric (%) | Method | AMZ→RDT | AMZ→FB | RDT→AMZ | RDT→FB | FB→AMZ | FB→RDT | Avg. |
|---|---|---|---|---|---|---|---|---|
| AUROC | GADT3 (pred. layer) | 59.09 | 94.80 | 80.76 | 94.98 | 74.39 | 64.25 | 78.05 |
| | GADT3 | 61.95 | 91.03 | 79.87 | 94.84 | 82.57 | 61.93 | 78.70 |
| AUPRC | GADT3 (pred. layer) | 4.48 | 28.27 | 29.17 | 31.98 | 14.53 | 6.73 | 19.19 |
| | GADT3 | 5.19 | 27.06 | 19.10 | 34.76 | 25.73 | 6.12 | 19.99 |

in better performance improvements. This suggests that when the source domain has higher homophily, the homophily-based test-time training loss function is more effective at guiding the model's adaptation to the target domain. For example, transferring from Reddit (higher homophily) to Facebook (lower homophily) achieves significantly better performance than the reverse direction. This finding provides practical guidance, indicating that domains with stronger homophilic structures may serve as better source domains for transfer learning in cross-domain anomaly detection tasks (see Figure 7).

## A.6  Evaluation using $\theta_{\text{pred}}$ for anomaly scoring

In addition to our ranking-based anomaly scoring, we evaluated an alternative setup where the frozen source-domain classifier $\theta_{\text{pred}}$ is used directly on the adapted target embeddings. Specifically, the predicted class probabilities were used to rank nodes by their likelihood of being anomalous. The same test-time training and early stopping procedures were applied. Performance results using this method are summarized in Table 7, allowing direct comparison with our primary ranking-based approach.

## A.7  Complexity analysis

The per-layer time complexity is $\mathcal{O}(|\mathcal{E}| \cdot d + |\mathcal{V}| \cdot d^2)$, where $|\mathcal{V}|$ and $|\mathcal{E}|$ are the number of nodes and edges, respectively, and $d$ is the hidden dimension of the projected space Li et al. (2025). Message passing is linear in the number of edges, and our attention mechanism—computed only over each node's neighbors—adds an extra term in $\mathcal{O}(|\mathcal{E}|)$, without incurring the $\mathcal{O}(|\mathcal{V}|^2)$ cost of dense similarity computation. Memory complexity

is $\mathcal{O}(|\mathcal{V}| \cdot d + |\mathcal{E}|)$ in the standard setting. During test-time adaptation (TTA), only the projection layer for the target domain is updated while the GNN and classifier remain frozen, reducing memory overhead for gradients. Runtime and memory usage with and without NSAW are provided in Table 8.

Table 8: Runtime and memory usage during TTA and additional overhead from NSAW.

| Dataset | TTA time(s) | TTA Memory (MB) | NSAW Time (s) | NSAW Memory (MB) |
|---------|-------------|-----------------|---------------|------------------|
| AMZ | 14.35 | 2264.17 | 0.01 | 0.21 |
| FB | 2.34 | 244.45 | 0.01 | 0.30 |
| RDT | 14.93 | 2630.33 | 0.02 | 0.14 |
| HTL | 10.41 | 3594.84 | 0.12 | 151.29 |
| RES | 16.29 | 3418.60 | 0.44 | 17.40 |
| AMZ-all | 52.13 | 3802.78 | 0.89 | 6.68 |
| YC-all | 50.47 | 4104.21 | 1.35 | 13.16 |
| TF | 190.62 | 7506.49 | 1.89 | 18.12 |

## A.8 Additional hyperparameter sensitivity analysis

Table 9 presents an ablation over key hyperparameters. We observe that setting $p = 40$, $\lambda_{\mathrm{reg}} = 0.1$, and $\lambda_s = 0.001$ consistently yields the best performance on both transfer tasks (AMZ→FB and FB→AMZ), indicating stable behavior across domains.

Table 9: Effect of hyperparameters on accuracy (%) for AMZ→FB and FB→AMZ. Best values in bold.

| Parameter | AMZ→FB | FB→AMZ |
|-----------|--------|--------|
| $p = 32$ | 90.1 | 81.6 |
| $\mathbf{p = 40}$ | **91.0** | **82.6** |
| $p = 64$ | 90.8 | 81.7 |
| $\lambda_{\mathrm{reg}} = 0.05$ | 90.2 | 81.0 |
| $\boldsymbol{\lambda_{\mathrm{reg}} = 0.1}$ | **91.0** | **82.6** |
| $\lambda_{\mathrm{reg}} = 0.2$ | 89.8 | 81.8 |
| $\lambda_s = 0.0005$ | 90.4 | 81.3 |
| $\boldsymbol{\lambda_s = 0.001}$ | **91.0** | **82.6** |
| $\lambda_s = 0.005$ | 90.5 | 81.7 |

## A.9 Component Contribution under Homophily and Feature Shift

We evaluate the contribution of each component of GADT3 under varying homophily levels and feature-space shifts. Specifically, we consider the AMZ→RDT transfer with high and low homophily (the latter synthetically adjusted to 0.3), as well as transfer pairs representing large (AMZ→RDT) and small (FB→AMZ) feature-space shifts. We report AUROC (%) when ablating each component individually in Table 10. As shown in the table, each component has a meaningful contribution to the overall performance under different settings. Please note that the cases with high homophily and large feature shift are identical.

## A.10 Proof of Proposition 1

The proof proceeds in two main steps. First, we establish by induction that the gradient dominance property, $\|g_N^{(k)}\| > \|g_A^{(k)}\|$, holds for all TTT steps. We use this property to show that the margin $\Delta_t^{(k)}$ is increasing.

Table 10: Component-wise ablation (AUROC %) under different homophily (0.9 vs. 0.3) and feature shift.

| Component | High Homophily (AMZ→RDT) | Low Homophily (AMZ→RDT) | Large Feature Shift (AMZ→RDT) | Small Feature Shift (FB→AMZ) |
|---|---|---|---|---|
| w/o NSAW | 61.35 | 56.27 | 61.35 | 81.63 |
| w/o Source | 61.54 | 57.11 | 61.54 | **84.76** |
| w/o ClassReg | 61.76 | 55.91 | 61.76 | 81.98 |
| **GADT3 (full)** | **61.95** | **58.30** | **61.95** | 82.57 |

We will prove by induction that $\|g_N^{(k)}\| > \|g_A^{(k)}\|$ for all $k \geq 0$. For the base case $k = 0$, we need to show that $\|g_N^{(0)}\| > \|g_A^{(0)}\|$. By Assumption 2, we have $\|g_C^{(0)} - g_C(\theta_s^*)\| \leq L\|\theta_t^{(0)} - \theta_s^*\|$. By the triangle inequality,

$$\|g_N^{(0)}\| - \|g_A^{(0)}\| \geq \|g_N(\theta_s^*)\| - \|g_A(\theta_s^*)\| - \|g_N^{(0)} - g_N(\theta_s^*)\| - \|g_A^{(0)} - g_A(\theta_s^*)\|.$$

From Assumption 1, $\|g_N(\theta_s^*)\| - \|g_A(\theta_s^*)\| = \varepsilon_0 > 0$. By choosing a sufficiently small $\delta$ such that $\|\theta_t^{(0)} - \theta_s^*\| < \delta$ (Assumption 3), we can make $\|g_N^{(0)} - g_N(\theta_s^*)\| < L\delta$ and $\|g_A^{(0)} - g_A(\theta_s^*)\| < L\delta$. Thus, by choosing $\delta$ small enough, we ensure $\|g_N^{(0)}\| - \|g_A^{(0)}\| > 0$. This establishes the base case.

For the inductive step, assume that for some $k \geq 0$, the hypothesis holds: $\|g_N^{(k)}\| > \|g_A^{(k)}\|$. We must show that this implies $\|g_N^{(k+1)}\| > \|g_A^{(k+1)}\|$. By definition, the *global TTT gradient* at step $k$ is

$$g_V(\theta_t^{(k)}) = \nabla_\theta \, \mathbb{E}_{v \in \mathcal{V}}[s_t(v; \theta)]|_{\theta = \theta_t^{(k)}} = r_N g_N(\theta_t^{(k)}) + r_A g_A(\theta_t^{(k)}),$$

where $r_N = |\mathcal{N}|/|\mathcal{V}|$ and $r_A = 1 - r_N$. By the nature of graph anomaly detection, the number of normal nodes is typically much larger than the number of anomalous nodes, so $r_N > r_A$. This implies that $(r_N - r_A) > 0$. Using the shorthand notations introduced earlier, this can be written as $g_V^{(k)} = r_N g_N^{(k)} + r_A g_A^{(k)}$.

The TTT update is $\theta_t^{(k+1)} = \theta_t^{(k)} + \eta g_V^{(k)}$. By Assumption 2, we have $\|g_C^{(k+1)} - g_C^{(k)}\| \leq L\|\theta_t^{(k+1)} - \theta_t^{(k)}\| = L\eta\|g_V^{(k)}\|$. Using the triangle inequality, we can bound the norms of the gradients at step $k + 1$:

$$\|g_N^{(k+1)}\| \geq \|g_N^{(k)}\| - \|g_N^{(k+1)} - g_N^{(k)}\| \geq \|g_N^{(k)}\| - L\eta\|g_V^{(k)}\|$$
$$\|g_A^{(k+1)}\| \leq \|g_A^{(k)}\| + \|g_A^{(k+1)} - g_A^{(k)}\| \leq \|g_A^{(k)}\| + L\eta\|g_V^{(k)}\|$$

Subtracting the second inequality from the first gives

$$\|g_N^{(k+1)}\| - \|g_A^{(k+1)}\| \geq \left(\|g_N^{(k)}\| - \|g_A^{(k)}\|\right) - 2L\eta\|g_V^{(k)}\|.$$

By the inductive hypothesis, $\|g_N^{(k)}\| > \|g_A^{(k)}\|$. Given the boundedness gradient assumption, we can choose a small enough $\eta$ such that the above term is positive, ensuring that the gradient dominance is preserved. This concludes the induction.

Now we can use this property to prove the proposition. A first-order Taylor expansion of $\Delta_t$ at $\theta_t^{(k)}$ gives

$$\Delta_t^{(k+1)} - \Delta_t^{(k)} = \eta \langle g_N^{(k)} - g_A^{(k)}, g_V^{(k)} \rangle + O(\eta^2).$$

Substituting this into the inner product, we get

$$\begin{aligned}
\langle g_N^{(k)} - g_A^{(k)}, r_N g_N^{(k)} + r_A g_A^{(k)} \rangle &= r_N \|g_N^{(k)}\|^2 - r_A \|g_A^{(k)}\|^2 + (r_A - r_N)\langle g_N^{(k)}, g_A^{(k)} \rangle \\
&\geq r_N \|g_N^{(k)}\|^2 - r_A \|g_A^{(k)}\|^2 - (r_N - r_A)\|g_N^{(k)}\|\|g_A^{(k)}\| \\
&> r_N \|g_N^{(k)}\|^2 - r_A \|g_N^{(k)}\|^2 - (r_N - r_A)\|g_N^{(k)}\|^2 \\
&= 0
\end{aligned}$$

For the first inequality, we apply the Cauchy-Schwarz inequality, which gives $|\langle g_N^{(k)}, g_A^{(k)} \rangle| \leq \|g_N^{(k)}\|\|g_A^{(k)}\|$. Moreover, from the inductive proof, we have $\|g_N^{(k)}\| > \|g_A^{(k)}\|$ hence the second inequality holds. Given that all terms in the final expansion are positive, the inner product term is guaranteed to be positive. For a sufficiently small $\eta$, the quadratic term $O(\eta^2)$ is negligible. The sign of $\Delta_t^{(k+1)} - \Delta_t^{(k)}$ is determined by the sign of the inner product term, which is positive. Thus, we have $\Delta_t^{(k+1)} > \Delta_t^{(k)}$ which concludes the proof.

### A.11 Homophily score patterns across datasets

Figures 8 and 9 show homophily score distributions for additional datasets.

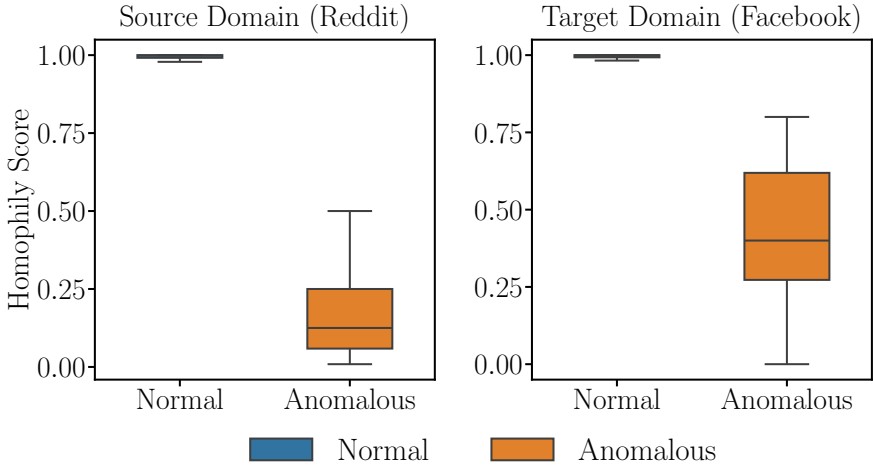

Figure 8: Homophily score distributions across domains (Reddit and Facebook)

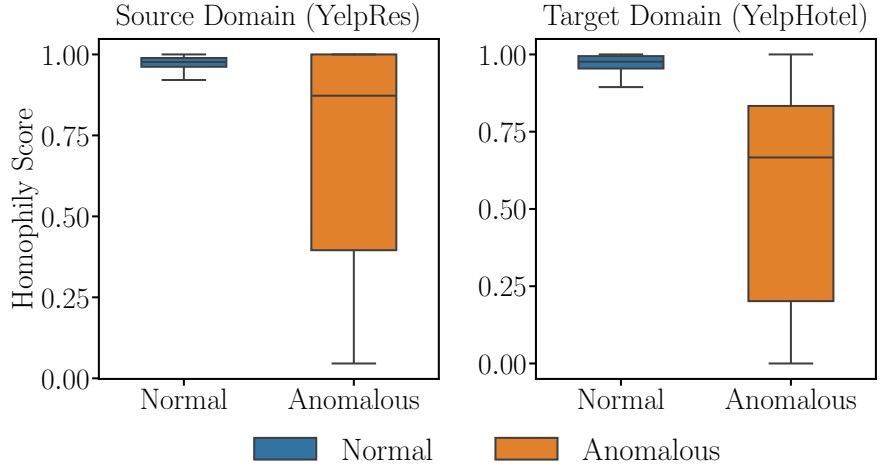

Figure 9: Homophily score distributions across domains (YelpRes and YelpHotel)

### A.12 Additional embedding visualizations

In Figure 10, we show 2D embeddings from our model for different sources and targets (similar to Figure 4).

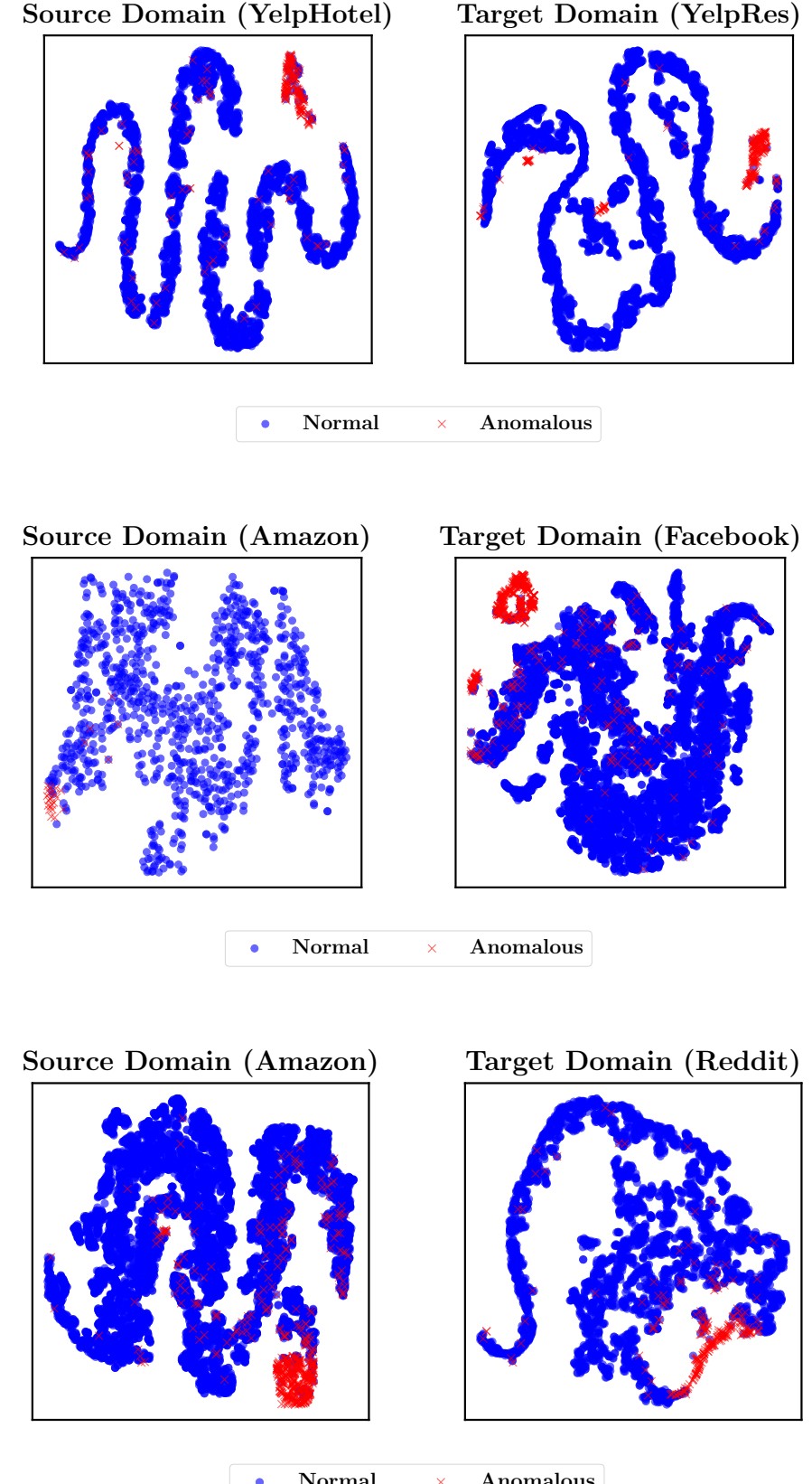

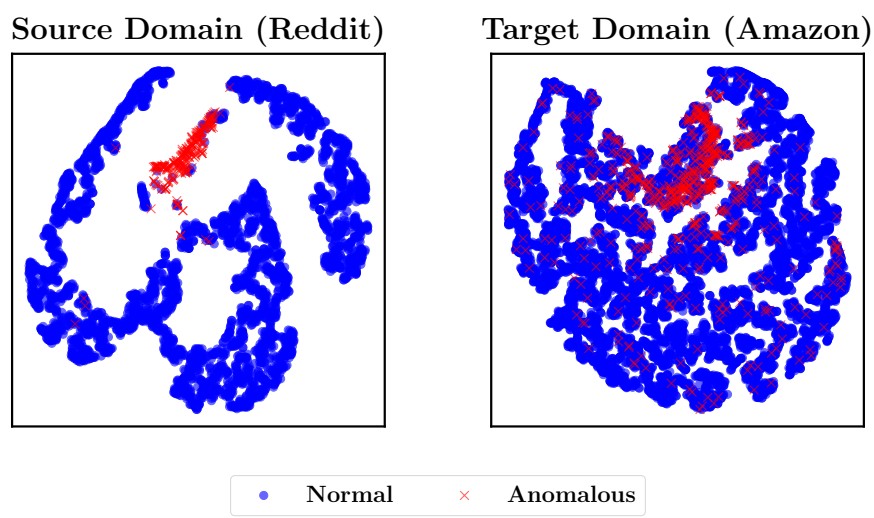

Figure 10: Learned 2D embeddings of various domain pairs as source/target data