# OpenReview forum: "Cross-Domain Graph Anomaly Detection via Test-Time Training with Homophily-Guided Self-Supervision"
_TMLR — Accepted by TMLR_

### Review · Reviewer_iFaJ · 2025-07-02

**Summary Of Contributions:**

This paper tackles the Graph Anomaly Detection (GAD) problem under the setting of unsupervised domain adaptation. The authors propose: 1) an unsupervised loss based on the empirical observation that normal nodes are more likely to be homophilic than anomalous ones. Specifically, the representations of adjacent nodes are aligned while the representations of disconnected nodes are pushed farther;
2) a symmetric attention technique to reduce the impact of anomalous nodes on the representations of the normal ones;
3) a class-aware regularization to handle the class imbalance issue. Experiments demonstrate the effectiveness of the method.

**Audience:**

Yes

**Claims And Evidence:**

Yes

**Requested Changes:**

1. Could you further clarify the motivation behind the design of the homophily loss and the robust symmetric attention?

**Strengths And Weaknesses:**

**Strengths:**
1. The proposed method shows strong empirical gains over previous methods.
2. The considered problem setting — unsupervised domain adaptation — is a very realistic setting that has high real-world applicability.

**Weakness:**

Although the empirical improvements are significant, however, the motivations for some of the algorithmic design are unclear for me.
    (1) **The homophily loss.**  According to your observation that "normal nodes are more similar to their neighbors than anomalous ones", shouldn't the homophily loss be only applied to normal nodes to better separate the normal nodes from the anomalous ones (further emphasize the homophily feature of the normal nodes to differentiate them from the anomalous nodes)? In Eq.5, the loss seems to be imposed on all nodes.
    (2) **The robust symmetric attention.** Why does reducing the information from abnormal neighbors help anomaly detection? Is it because we need to encourage the difference between the learned representations for normal nodes and anomalous ones?

---

> ### Author Response · Authors · 2025-08-13
> **Response to reviewer-iFaJ**
>
> We would like to thank the reviewer for insightful comments. Below we address the motivation behind the homophily loss and the robust symmetric attention mechanism.
>
> **Homophily loss**: Thank you for the insightful comment. We apply the homophily loss to all connected nodes due to the unsupervised setting, as we do not know which nodes are normal. This loss encourages all connected nodes to become similar, even anomalous pairs. However, because our datasets (and most real-world datasets) are homophilic, there are significantly fewer anomalous pairs than normal.
>
> A regularization term pushes non-connected nodes apart to preserve structural contrast.
>
> Our attention mechanism (NSAW) tries to assign lower weights to anomalous-normal (non-homophilic) pairs to further differentiate their representations.
>
> We have modified Section 4.2 as follows:
>
> *“Since node labels are unavailable during training, the homophily loss is applied to all connected node pairs. This may include anomalous pairs, but under our homophily assumption, most edges still connect similar (normal) nodes, making the signal reliable in practice.”*
>
> **Robust Symmetric Attention:** Thank you for pointing this out. The homophily loss may mistakenly enforce similarity between some normal-anomaly pairs if they are connected. To address this, we introduce NSAW, which learns attention weights and weakens edges likely to connect anomalies and normals. Also, the symmetric minimum property ensures that if attention is high in one direction (e.g., anomaly → normal) but low in the other, the lower value is used, preventing assigning high values to noisy edges.
>
> We have modified Section 4.3 as follows:
>
>
> *“Specifically, by downweighting abnormal-to-normal connections, NSAW reduces the influence of structurally inconsistent neighbors. This preserves the structure of normal regions and encourages more distinct representations for anomalies.”*
>
> Together, the homophily loss promotes similarity within the local neighborhood, and NSAW prevents noisy edges (e.g., normal–anomaly, anomaly-normal) from distorting that alignment. This combination enables robust adaptation without requiring target labels.

---

### Review · Reviewer_Ga9t · 2025-07-03

**Summary Of Contributions:**

The paper introduces **GADT3**, the first *test-time training (TTT)* framework specifically designed for *cross-domain graph anomaly detection*. GADT3 combines supervised learning on source graphs and homophily-guided self-supervision at test time to adapt to out-of-distribution and heterogeneous target graphs without requiring target labels or source data access during adaptation. Key innovations include a *homophily-based affinity score* to capture domain-invariant anomaly signals, *Normal Structure-preserved Attention Weighting (NSAW)* to dynamically suppress anomalous influence during message passing, *domain-specific encoders* to handle differing feature spaces, and *class-aware regularization* to address class imbalance. Extensive experiments across diverse datasets demonstrate that GADT3 consistently outperforms existing domain adaptation and anomaly detection baselines.

**Audience:**

Yes

**Broader Impact Concerns:**

None.

**Claims And Evidence:**

Yes

**Requested Changes:**

See weaknesses above.

**Strengths And Weaknesses:**

Strengths:

* S1: Proposes the first test-time training (TTT) framework specifically tailored for cross-domain graph anomaly detection, addressing an underexplored problem.
* S2: Introduces several well-motivated innovations (homophily-based self-supervision, NSAW, domain-specific encoders, and class-aware regularization), each targeting a specific challenge in cross-domain GAD.
* S3: The experiments are extensive, covering multiple cross-domain scenarios (heterogeneous and homogeneous features), large-scale graphs, ablations, and sensitivity analyses to homophily.
* S4: GADT3 consistently outperforms state-of-the-art baselines, demonstrating clear and substantial improvements in AUROC and AUPRC.
* S5:  The paper is generally well-structured, with an intuitive overview figure (Fig. 2) and consistent notation.

Weaknesses:

* W1: The method relies heavily on the assumption that normal nodes have higher homophily, which may not hold in heterophilic or less structured graphs; although this is acknowledged, it remains a limitation.
* W2: The paper lacks formal analysis or guarantees about adaptation performance (e.g., bounds relating source-target similarity to generalization).
* W3: Since homophily is a core assumption, it would have been informative to compare against methods that do not assume homophily.
* W4: While the ablations are useful, the analysis could be more granular—for example, quantifying the contribution of each component across varying homophily levels or feature-space shifts separately.
* W5: Although large graphs are tested, the computational cost (e.g., runtime, memory) of test-time adaptation and NSAW isn’t thoroughly discussed.
* W6: All baselines use a projection head (to match features) by the authors; it is unclear how this impacts their performance.
* W7: Lacking discussion about how hyperparameters were chosen. You list them directly under Table 1 without discussing how you obtained those particular values.

---

> ### Author Response · Authors · 2025-08-13
> **Response to reviewer-Ga9t (part 1)**
>
> We would like to thank the reviewer for the insightful comments. Below, we address your concerns and provide clarification on specific points.
>
> **W1: Homophily assumption.**
> We agree and discuss this in Sections 6 and 7. To further address this, we have conducted robustness experiments with synthetically rewired graphs to reduce homophily (Table 4). Results show GADT3 maintains strong performance when homophily is moderate (≥0.5), and degrades gradually when the graphs become more heterophilic. This is expected, as our SSL loss relies on local similarity to guide adaptation.
>
> However, GADT3 can be extended to such settings by replacing the current SSL loss with a loss more suited for heterophily. Recent works have proposed GAD for heterophilic graphs [1].
>
> [1] Addressing heterophily in graph anomaly detection: A perspective of graph spectrum. WWW’23
>
> We’ve modified Section 2.2 as follows:
>
> *“While our method is motivated by the homophily assumption, recent studies such as Gao et al. have explored anomaly detection in **heterophilic graphs**, highlighting the importance of developing models robust to varying structural patterns. Our approach can be extended to heterophilic graphs by replacing the current SSL loss with a loss more suited for heterophily, and by using heterophily-aware GNNs.”*
>
> **W2: guarantees about adaptation performance.**
> We thank the reviewer for pointing out the lack of formal guarantees in our method. In response, we have added a formal result **(Proposition~1)** in Section 4.6 (with detailed proof in Appendix A.10), showing that our  method is guaranteed to monotonically improve the separation between normal and anomalous nodes in the target graph during adaptation.
>
> **Proposition 1 (short form):**
> Let   $
> \Delta_t^{(k)} = E_{v_N \in N}[S_t(v_N; \theta_t^{(k)})] - E_{v_A \in A}[S_t(v_A; \theta_t^{(k)})]$
> be the anomaly separation margin at step $k$. If $\theta_t^{(0)}$ is initialized close to $\theta_s^*$, then the following holds:
>
> $$
> \Delta_t^{(k+1)} > \Delta_t^{(k)}, \quad \forall k \ge 0.
> $$
>
>
> In other words, each TTT update step increases the difference in homophily scores between the two classes, ensuring that the adaptation consistently moves the model toward better anomaly discrimination rather than degrading it. This provides theoretical backing for our SSL loss design and confirms that the optimization process steadily enhances anomaly separability.
>
> The proof relies on three assumptions: (1) the source encoder produces stronger gradient signals from normal nodes than anomalous ones (at the source optimum), (2) class-conditional gradients vary smoothly with the encoder parameters, and (3) target encoder initialization is close to the source optimum.
>
>
> **W3: comparison against methods that do not assume homophily.**
> We appreciate the reviewer’s observation. While GADT3 assumes higher homophily for normal nodes, most baselines, such as DOMINANT, ComGA, and GTrans, are homophily-agnostic. Among our baselines, only TAM incorporates homophily. Despite this, GADT3 consistently outperforms all of them, even on less homophilic graphs.
>
> This suggests that our homophily-guided self-supervision offers a valuable inductive bias in cross-domain settings.
>
> We modified Section 5.1 as follows:
>
> *“While GADT3 assumes that normal nodes are more homophilic than anomalies, most baselines (e.g., DOMINANT, CoLA, SL-GAD, GTrans) are homophily-agnostic. Yet, GADT3 outperforms them, indicating that homophily-based self-supervision provides a useful inductive bias for cross-domain anomaly detection.”*
>
> **W4: Lack of more granular ablation studies.**
> We have done new experiments under two levels of homophily and feature shift. Table 10 (Appendix A.9) reports AUROC (%) while removing one component at a time. We select **AMZ->RDT** to represent a large feature space shift due to the semantic mismatch between product-related and textual features.  ​​In contrast, **FB->AMZ** represents a **smaller shift**, as both datasets involve user behavior features, leading to more aligned feature spaces.
>
>
> | Component                      | High homophily (AMZ→RDT) | Low homophily  (AMZ→RDT) | Large feature shift (AMZ→RDT) | Small feature shift (FB→AMZ) |
> | ------------------------------ | ------------------------ | ----------------------------- | ----------------------------- | ---------------------------- |
> | w/o NSAW                       | 61.35                    | 56.27                         | 61.35                         | 81.63                        |
> | w/o Source                     | 61.54                    | 57.11                         | 61.54                         | **84.76**                    |
> | w/o ClassReg | 61.76                    | 55.91                         | 61.76                         | 81.98                        |
> | GADT3 (full)               | **61.95**                | **58.30**                     | **61.95**                     | 82.57                        |

---

> ### Author Response · Authors · 2025-08-13
> **Response to reviewer-Ga9t (part 2)**
>
> **W5: Discussion on computational cost.**
> We thank the reviewer for pointing this out. Please refer to the response to W2 of reviewer yEmE for detailed analyses.
>
>
> **W6: Clarification about use of projection head by baselines.**
> Thank you for pointing this out. We added the projection head to all baselines only to handle different feature spaces across source/target. To ensure fairness, we used a simple single linear layer and applied it identically across all methods. Adding this might slightly affect the performance of the baselines; however, our method outperformed the baselines even in the homogeneous settings where we did not add any projection head to the baselines.
>
> We modified the first paragraph of Section 5.2 as follows:
>
> *“We incorporated a projection head into all baseline models to address mismatches in feature sets between source and target domains. This ensures a fair comparison by allowing each method to handle discrepancies in input feature spaces. In homogeneous settings, where source and target graphs share the same feature space, no projection head was used.”*
>
> **W7: Discussion about hyperparameters.**
> Thank you for pointing this out. We had already included a sensitivity analysis using a grid search for the main hyperparameters ($\lambda$ and $\alpha$) in Appendix A.3, but regretfully did not cite it clearly in the paper. We have now modified this in the paper in Section 5.1 (Implementation details).
>
> Additionally, we have added additional tuning results for $\lambda_s$, $\lambda_{reg}$, and embedding dimension $p$. These were selected using a small labeled validation set from the source domain (same method as A.3). We have added these results in Appendix A.8. The AUROC (%) results are as follows:
>
>
> | Parameter                 |   AMZ→FB |   FB→AMZ |
> | ------------------------- | -------: | -------: |
> | *p* = 32                  |     90.1 |     81.6 |
> | **p = 40**                | **91.0** | **82.6** |
> | *p* = 64                  |     90.8 |     81.7 |
> |$\lambda_{reg}$ = 0.05    |     90.2 |     81.0 |
> | **$\lambda_{reg}$ = 0.1** | **91.0** | **82.6** |
> | $\lambda_{reg}$= 0.2     |     89.8 |     81.8 |
> | $\lambda_s$ = 0.0005    |     90.4 |     81.3 |
> | **$\lambda_s$ = 0.001** | **91.0** | **82.6** |
> | $\lambda_s$ = 0.005     |     90.5 |     81.7 |

---

### Review · Reviewer_yEmE · 2025-08-03

**Summary Of Contributions:**

This paper introduces GADT3, a novel test-time training (TTT) framework for cross-domain graph anomaly detection (GAD) to address challenges like distribution shifts and heterogeneous feature spaces between source and target domains. GADT3 combines supervised and self-supervised learning during training, leveraging a homophily-based affinity score (capturing domain-invariant anomaly patterns) for self-supervision, and adapts to target domains at test time using only self-supervised learning. Key innovations include Normal Structure-preserved Attention Weighting (NSAW) to suppress anomalous node influences, domain-specific encoders for heterogeneous features, and class-aware regularization for class imbalance. Extensive experiments across multiple cross-domain datasets show GADT3 outperforms baselines, with average improvements over 8.2% in AUROC and AUPRC.

**Audience:**

Yes

**Claims And Evidence:**

No

**Requested Changes:**

Expand baseline comparisons with state-of-the-art post-2023 methods: The current baselines do not include the latest cross-domain GAD techniques published after 2023, which may undermine the claim of "state-of-the-art" performance.

**Strengths And Weaknesses:**

Strengths：
1. GADT3 is the first to apply test-time training to cross-domain GAD, enabling adaptation to target domains without requiring source data or target labels at test time, which enhances practicality for real-world scenarios with privacy constraints.
2.  By leveraging the domain-invariant pattern that normal nodes exhibit higher homophily than anomalous ones, the self-supervised loss aligns closely with the anomaly detection objective, improving cross-domain generalization.
3. NSAW (learnable attention weights to suppress anomalous influences) and class-aware regularization (addressing extreme imbalance) are tailored to GAD challenges, demonstrating strong design choices supported by ablation studies.

Weaknesses：
1. Performance degrades in low-homophily target domains (homophily < 0.5), limiting generalization to highly heterophilic graphs where normal and anomalous nodes show similar connectivity patterns.
2. While tested on large-scale graphs, the paper provides little analysis on computational efficiency or scalability trade-offs, which is critical for deployment on extremely large real-world graphs.
3.  Although compared to several methods, GADT3 is not evaluated against the latest cross-domain GAD techniques beyond 2023, potentially limiting the completeness of its superiority claims.

---

> ### Author Response · Authors · 2025-08-13
> **Response to reviewer-yEmE**
>
> We would like to thank the reviewer for the insightful comments. Below, we address your concerns and provide clarification on specific points.
>
> **W1: Generalization to heterophilic graphs.**
> We agree and acknowledge this in Sections 6 and 7. Our SSL loss relies on local similarity to guide adaptation.
>
> However, GADT3 can be extended to such settings by replacing the current SSL loss with a loss more suited for heterophily, and by using heterophily-aware GNNs. Recent works have proposed GAD for heterophilic graphs [1].
>
> [1] Addressing heterophily in graph anomaly detection: A perspective of graph spectrum. WWW’23
>
> We’ve added this to the paper in Section 2.2:
> *“While our method is motivated by the homophily assumption, recent studies such as Gao et al. have explored anomaly detection in **heterophilic graphs**, highlighting the importance of developing models robust to varying structural patterns.”*
>
> **W2: Lack of analysis on computational efficiency or scalability trade-offs.**
> We thank the reviewer for raising this point. We have now added both theoretical and empirical analysis of runtime and memory complexity to Appendix A.7 in detail. The time complexity is $\mathcal{O}(|\mathcal{E}| \cdot d + |\mathcal{V}| \cdot d^2)$ per layer. The empirical results (Table~8) show that the NSAW attention mechanism introduces minimal overhead during TTA, typically under 2s and 20 MB for most datasets, even large graphs. This confirms the method’s scalability across diverse graph sizes. As shown in the table, GADT3 offers faster adaptation runtime compared to TAM (SOTA) and moderate memory usage.
>
>
> | Dataset | TTA time (s) | TTA memory (MB) | NSAW time (s) | NSAW memory (MB) |
> | ------- | --------------: | -------------------------: | --------------------: | -------------------: |
> | AMZ     |           14.35 |                    2264.17 |                  0.01 |                 0.21 |
> | FB      |            2.34 |                     244.45 |                  0.01 |                 0.30 |
> | RDT     |           14.93 |                    2630.33 |                  0.02 |                 0.14 |
> | HTL     |           10.41 |                    3594.84 |                  0.12 |               151.29 |
> | RES     |           16.29 |                    3418.60 |                  0.44 |                17.40 |
> | AMZ-all |           52.13 |                    3802.78 |                  0.89 |                 6.68 |
> | YC-all  |           50.47 |                    4104.21 |                  1.35 |                13.16 |
> | TF      |          190.62 |                    7506.49 |                  1.89 |                18.12 |
>
>
> **W3: Comparison against more recent baselines**
> We thank the reviewer for highlighting this point:
> 1. We’ve already cited GAD methods from 2024 (Qiao & Pang, 2024) and (Chen et al, 2024). Although both are recent and strong homophily-based anomaly detectors, they do not support cross-domain generalization. Our work extends this direction by addressing the harder CD-GAD setting under TTA.
>
> 2. We have included **ARC (NeurIPS 2024)**, a recent cross-domain GAD method that performs **in-context adaptation at inference** using a few normal nodes from the target domain. ARC reports results on AMZ, FB, and RDT, which overlap with our evaluation benchmarks. While this baseline assumes access to normal target examples during inference, our method adapts using the **entire target**, without requiring any supervision. We will include results for full datasets in the paper. Performance comparisons (AUC) are provided in the Table below:
>
>
> | Dataset |   ARC | GADT3 (Ours) |
> | ------- | ----: | -----------: |
> | AMZ     | 80.67 |        82.57 |
> | FB      | 67.56 |        94.84 |
> | RDT     | 60.04 |        61.95 |
>
> 3.  We have included **CDFS-GAD (2024)**, a semi-supervised CD-GAD method. Below, we compare the performance (AUC/AP) with our method:
>
>
> | Method                                | HTL → RES  | RES → HTL  |
> | ------------------------------------- | -------------------: | -------------------: |
> | CDFS-GAD                              |          99.2 / 90.3 |          79.8 / 26.8 |
> | **GADT3 (Ours)**  |        94.14 / 39.75 |        90.71 / 42.90 |
>
>
> While CDFS-GAD performs better on the HTL → RES direction under its few-shot setting, **GADT3 outperforms significantly on the reverse direction (RES -> HTL)**, especially in AP, and does so in a fully unsupervised setting. This highlights the robustness of our model in the absence of target labels. We will add the results for all datasets in our paper. We’ve cited this method in our related work as follows:
>
> *“Other recent works like CDFS-GAD \citep{chen2024towards} address cross-domain node anomaly detection in a few-shot setting, where a small set of labeled anomalies from the target graph is available. ARC \citep{liu2024arc} proposes an in-context learning approach for cross-domain GAD, which adapts at inference using a few normal nodes from the target graph.”*

---

### Decision · Action_Editor_i7bE · 2025-09-21

**Recommendation:** Accept as is

**Audience:**

Yes

**Audience Explanation:**

This paper makes a solid contribution to cross-domain graph anomaly detection by introducing the first test-time training approach for this setting. The method demonstrates strong empirical results across diverse datasets and provides novel technical innovations that are well-motivated and experimentally validated. While there are limitations regarding the homophily assumption, the authors acknowledge these constraints, and the work opens up promising research directions. The comprehensive experimental evaluation, updated theoretical analysis, and practical applicability make this a valuable contribution to the field.

**Claims And Evidence:**

Yes

**Claims Explanation:**

This paper introduces GADT3, a novel test-time training (TTT) framework for cross-domain graph anomaly detection. The method combines supervised and self-supervised learning during training while adapting to new domains at test time using only self-supervised learning. The approach leverages a homophily-based affinity score to capture domain-invariant properties of anomalies and introduces several technical innovations, including Normal Structure-preserved Attention Weighting (NSAW), domain-specific encoders, and class-aware regularization.

Reviewers agree that GADT3 represents the first application of test-time training to cross-domain graph anomaly detection, addressing an important and underexplored problem in the field. Initial concerns from the reviewers include:

- The method's reliance on the homophily assumption, which limits its applicability to heterophilic graphs (homophily < 0.5)
- Lack of empirical comparisons with more recent works in the literature post 2023
- Lack of analysis on the computational efficiency of the proposed method

The authors provided comprehensive responses to address the above concerns and all the reviewers are satisfied with the updates. I recommend acceptance as is.